



# Recent above-ground biomass changes in central Chukotka (Russian Far East) using field sampling and Landsat satellite data

Iuliia Shevtsova[1,2], Ulrike Herzschuh[1,2,3], Birgit Heim[1], Luise Schulte[1,2], Simone Stünzi[1,6], Luidmila A. Pestryakova[4], Evgeniy S. Zakharov[4,5], and Stefan Kruse[1]

[1]Polar Terrestrial Environmental Systems, Alfred Wegener Institute (AWI), Helmholtz Centre for Polar and Marine Research, Potsdam, 14473, Germany
[2]Institute of Biochemistry and Biology, University of Potsdam, Potsdam, 14476, Germany
[3]Institute of Environmental Sciences and Geography, University of Potsdam, Potsdam, 14476, Germany
[4]Institute of Natural Sciences, North-Eastern Federal University of Yakutsk, Yakutsk, 677000, Russia
[5]Institute for Biological Problems of the Cryolithozone, Russian Academy of Sciences, Siberian branch, Yakutsk, 677000, Russia
[6]Geography Department, Humboldt-Universität zu Berlin, Unter den Linden 6, 10099, Berlin, Germany

*Correspondence to*: Iuliia Shevtsova (iuliia.shevtsova@awi.de), Stefan Kruse (stefan.kruse@awi.de)

**Abstract.** Upscaling plant biomass distribution and dynamics is essential for estimating carbon stocks and carbon balance. In this respect, the Russian Far East is among the least investigated subarctic regions despite its known vegetation sensitivity to ongoing warming. We representatively harvested above-ground biomass (AGB, separated by dominant taxa) at 40 sampling plots in central Chukotka. We used ordination to relate field-based taxa projective cover and Landsat-derived vegetation indices. A general additive model was used to link the ordination scores to AGB. We then mapped AGB for paired Landsat-derived time-slices (i.e. 2000/2001/2002 and 2016/2017), in four study regions covering a wide vegetation gradient from closed-canopy larch forests to barren alpine tundra. We provide AGB estimates and changes in AGB that were previously lacking for central Chukotka at a high spatial resolution and a detailed description of taxonomical contributions. Generally, AGB in the study region ranges from 0 to 16 kg m$^{-2}$, with Cajander larch providing the highest contribution. Comparison of changes in AGB within the investigated period shows that the greatest changes (up to 1.25 kg m$^{-2}$ yr$^{-1}$) occurred in the northern taiga and in areas where land cover changed to larch closed-canopy forest. As well as the notable changes, increases in AGB also occur within the land cover classes. Our estimations indicate a general increase in total AGB throughout the investigated tundra-taiga and northern taiga, whereas the tundra showed no evidence of change in AGB.

## 1 Introduction

Estimated global mean surface temperature has increased by 0.87 °C since pre-industrial times and continues to rise (IPCC, 2019). The Arctic is warming two to three times faster than the global annual average. Here, vast amounts of terrestrial carbon are stored in the soil organic matter and living plant





biomass (McGuaer et al., 2009; ACIA, 2005) and, therefore, changes in the carbon cycle potentially affected by climate change are a central issue. In the course of global warming, positive feedbacks can be observed: for example, encroachment of deep-rooted vegetation due to shrubification can lead to deeper

carbon deposition and act as a potential carbon sink (Jobbágy and Jackson, 2000). Therefore, estimation of above-ground biomass (AGB) stocks and detailed knowledge about the individual taxa contributing to it is of prime interest to understand whether northernmost forests and tundra also change in biomass in analogy to the widespread observed shrubification. This information is essential for modelling terrestrial carbon cycling in vulnerable high-latitude ecosystems and will help predict future carbon dynamics that

may accelerate or slow down future warming.

Detailed (species/taxa level) estimation of AGB can provide more valuable information on ecosystem's functioning and its development than AGB estimates at a plant functional type (PFT) level. For example, a loss of specific species from one PFT can be replaced by taxa from another PFT in response to climate change even though total AGB production remains similar (Bret-Harte et al, 2008). Thus, the change in

AGB between PFTs can be caused by changing species contributions within PFTs. However, many studies of arctic and subarctic regions present AGB state or change at a PFT level (Räsänen et al, 2018; Berner et al, 2018; Webb et al, 2017; Walker et al, 2003). Some focus only on shrub biomass of one or more species (Vankoughnett and Grogan, 2015; Berner et al, 2014), others on tree biomass (Berner et al, 2012), or on species and PFT AGB of a one specific community (e.g. Hudson and Henry, 2009). Rarely,

a study presents results of AGB on a PFT level despite sampling methods that suggest a division by species in the field (Maslov et al, 2016; Chen et al, 2009). Very seldom, AGB is presented at a species/taxa level (e.g. Shaver and Chapin, 1991). In consequence, only a few estimations of species or taxon-specific AGB are available to assess species/taxa contributions.

Whereas for some Arctic regions in North America, AGB state and change have been well studied (e.g.

Canada, Hudson, 2009), the Russian Far East has received less attention and AGB has never been investigated in the vast areas of central Chukotka, which is our study region. The very few existing circumpolar AGB estimations that also cover these areas (Raynolds et al., 2011; Santoro and Cartus, 2019) have a coarse spatial resolution (1 km and 100 m respectively) and, therefore, show only the general AGB gradient of lowest in tundra to highest in taiga. Similarly, the circumpolar estimation of Epstein et





al. (2012) covers AGB change until 2010 and shows only a general zonal pattern of change. In consequence, it remains unknown how the landscape of central Chukotka, with its characteristic treeline formed by needle-leaf deciduous trees, mountainous terrain, and high diversity of vegetation communities, responds to climate warming in terms of terrestrial carbon stocks.

For vegetation and ABG investigations the remote sensing index Normalised Difference Vegetation Index
(NDVI) is often used. It incorporates information from red and near infra-red regions of the light spectrum that reflect plant biomass of various ecological systems (Pettorelli, 2006). In the Arctic and subarctic regions remote-sensing algorithms based on satellite derived NDVI and field measurements were used to predict the total and exclusively shrub AGB in Alaska (Epstein et al, 2008; Berner et al, 2018) and for Cajander larch in north-eastern Siberia (Berner et al, 2012). Some studies have used very high spatial-
resolution imagery (Räsanen et al., 2018) and hyperspectral field spectrometry for AGB investigations in north-western and northern Siberia and Alaska (Bratsch, 2017), that enable spatially restricted studies on estimations of local AGB. To capture more precisely the AGB variability in our study region, Shevtsova et al. (2020) established a redundancy analysis model (RDA) that incorporates Landsat NDVI, Normalised Difference Water Index (NDWI), and Normalised Difference Snow Index (NDSI). This
model, together with the extensive Landsat satellite data archive, makes it possible to assess the strength and direction of AGB changes in central Chukotka over the last decades.

We used available Landsat satellite data and field data from a 2018 expedition in a statistical model for AGB mapping. The aim was to provide an estimation of AGB stocks and their change between paired time points (2000/2001/2002 to 2016/2017) at four focus areas along a tundra–taiga gradient, in central
Chukotka. Our first objective was to reconstruct the AGB of each sampling plot using individual plant biomass samples and their corresponding distribution within these plots. The second objective was to upscale AGB in the focus areas for the most recent time covered by Landsat-8 satellite data via statistical modelling. Finally, the third objective was to apply the developed upscaling approach to the oldest available good quality Landsat-7 acquisitions to investigate AGB changes in the focus areas.





## 2 Materials and methods

### 2.1 Study region and field surveys

Our study covers six areas of central Chukotka, Russian Far East (Fig. 1). Four of them (16-KP-01, 16-KP-02, 16-KP-03, 16-KP-04) are our focus areas for biomass mapping and previous vegetation investigations (Shevtsova et al., 2020a); two further areas (18-BIL-01, 18-BIL-02) are supplementary and were investigated for representative AGB sampling. All investigated areas are underlain by continuous permafrost and all four focus areas are mountainous.

During the expedition "Chukotka 2018" in July 2018, we inventoried 40 sample plots (Fig. 1; Biskaborn et al., 2019): five sample plots in treeless tundra (16-KP-04), 27 sample plots in the tundra–taiga ecotone (16-KP-01), and eight sample plots in northern taiga (18-BIL-01, 18-BIL-02). Fifteen-metre radius sampling plots were demarcated in the most homogeneous locations. Heterogeneity was accommodated by roughly assorting vegetation into two to three vegetation types per sampling plot. Within each area of roughly estimated vegetation types we selected three representative 2 x 2 m subplots for ground-layer foliage projective cover assessment. In these subplots, a 50 x 50 cm area was selected for ground-layer ABG harvesting (major taxa and others), as well as a 10 x 10 cm area for moss and lichen biomass harvesting (Fig. 2). AGB was sampled in 38 sample plots of the 40 inventoried.

All biomass samples were weighed fresh in the field. In general, biomass samples with a weight of more than 15 g were subsampled to reduce the volume of biomass as there were limits to what was logistically possible to transport to the laboratory for drying. All samples were oven dried (60 °C, 24 h for ground-layer and moss and lichen samples, 48 h for shrub and tree branch samples, up to one week for tree stem discs) and weighed again.

Our 2018 vegetation and biomass sampling plots were consistently placed in similar vegetation communities to those investigated in 2016. Only tall dense *Alnus viridis* ssp. *fruticosa* (Rupr.) Nyman (hereafter *Alnus fruticosa*) shrub associations were not sampled during the expedition in 2018, which is a rare type of vegetation community that only occurs in a few places in the area of interest. Additionally, we sampled the vegetation at an old fire scar, mostly consisting of patches of tall non-creeping *Salix* spp. shrubs with graminoids and dead, upright tree stems of *Larix cajanderi* Mayr.



The sampling protocols for projective cover and AGB sampling are different for 1) trees (all *Larix cajanderi*), 2) non-creeping shrubs (*Salix* spp., *Alnus fruticosa*, *Pinus pumila* (Pall.) Regel), and 3) ground-layer plants (including creeping shrubs, herbs, mosses, and lichens).

Tree cover and heights of all trees were visually estimated in the 15 m radius plot after training with a clinometer (SUUNTO, Finland). Detailed parameters of ten trees per 15 m radius plot were recorded: height, crown diameter, crown start, stem perimeter at basal and at 1.3 m height, and vitality. We aimed to representatively sample at least three (tall, medium, low) of these trees for ABG. Samples included, if available, needle biomass, one small living branch, one medium-sized living branch, one big living

branch, one dead branch, and ideally three stem discs (basal height at 0 cm, breast-height at 130 cm, and 260 cm height). From the 107 trees sampled, 53 trees were fully sampled, 41 trees were sampled only from the tree trunk, and 13 trees only from branches and needles. Stem biomass was reconstructed using allometric equations (Appendix A) based on the assumption of a cone-shaped tree form. Using exponential models, we were able to reconstruct total and partial (wood, needle) ABG of all trees

(separately for dead and living trees) in each 15 m radius plot. We converted our AGB estimates into averages of kg m$^{-2}$ for each 15 m radius plot.

Non-creeping shrub cover was estimated in the 15 m radius plot. If present, three representative shrub individuals from each species were sampled for AGB: leaf/needle and branch. The average total and partial AGB from representative shrubs were then converted to kg m$^{-2}$ for each sample plot (Appendix

A).

Ground-layer vegetation cover was estimated in 2 x 2 m representative subplots. AGB of ground-layer plants was estimated by harvesting 50 x 50 cm subplots; AGB of mosses and lichens by harvesting 10 x 10 cm subplots. By accounting for the vegetation types within each 15 m radius plot, the total average ABG of each sampled taxon was estimated in kg m$^{-2}$ per sample plot (details in Appendix A).

All AGB estimations (total and per taxon) were analysed in four land-cover classes (1: larch closed-canopy forest, 2: forest tundra and shrub tundra, 3: graminoid tundra, 4: prostrate herb tundra and barren areas (Shevtsova et al., 2020a)) and are reported by their median with interquartile range (IQR) as a measurement of statistical dispersion.





## 2.2 Above-ground biomass upscaling and change derivation

A redundancy analysis (RDA) model was built with foliage projective cover of 36 taxa from the 2016
expedition sample plots as dependent variables and Landsat spectral indices (Normalised Difference
Vegetation Index (NDVI), Normalised Difference Water Index (NDWI), Normalised Difference Snow
Index (NDSI)) as predictors (Shevtsova et al., 2020a). We used the RDA model to predict RDA scores
for the 40 new sample plots of the 2018 expedition. Foliage projective cover of the new sample plots

covered the same taxonomical resolution and was standardised by applying a Hellinger transformation
(Legendre and Legendre, 2012). Every position in the ordination space describes a specific vegetation
composition with a specific coverage, as well as a combination of Landsat spectral indices associated
with it. Using the RDA scores, we assigned sample plots from the 2018 expedition to the four established
land-cover classes using k-means classification: (1) larch closed-canopy forest, (2) forest tundra and shrub

tundra, (3) graminoid tundra, (4) prostrate herb tundra and barren areas (Shevtsova et al., 2020a).
For predicting the total AGB for the 2018 sample plots, the RDA scores of the two first axes were used
to build a generalised additive model (GAM, R package "mgcv") using Eq. (1).

$$Total\ AGB = \text{RDA1} + \text{s(RDA1, RDA2)}\ , \tag{1}$$

where RDA1 and RDA2 are the ordination scores of the first and second axes, respectively, of the 2018

expedition data from sample plots where ABG was sampled, and s is a smooth monotonic function. The
parameterised GAM was subsequently used to estimate the total AGB for the four focus areas based on
the RDA-scores of Landsat spectral indices (Table 1). Specifically, for each focus area the AGB was
mapped for each of two time points: recent (2016 or 2017) and historical (2000, 2001 or 2002). From
AGB maps with 15–16 years difference covering the same focus area, AGB change maps were produced.

The state and any change of AGB were estimated within and between land-cover classes for land-cover
state and change maps (Shevtsova et al., 2020a). All final estimations of AGB state are presented in kg
m$^{-2}$ as median with IQR.

All analyses were done in R (R Core Team, 2017) using the packages "vegan" version 2.5-4 (Oksanen et
al., 2019), "raster" version 2.6-7 (Hijmans, 2017), "mgcv" (Wood, 2011), "sp" (Pebesma and Bivand,

2005), "factoextra" version 1.0.5.999 (Kassambra and Mundt, 2017), and "ggplot2" (Wickham, 2016).



## 3 Results

### 3.1 Vegetation composition and above-ground biomass

In situ projective cover data of all 2018 expedition vegetation sample plots are described in Shevtsova et al. (2020b). The main vegetation communities of the study region assessed were: (1) barren areas, covered only by rock lichens; different vegetation associations of the open tundra such as (2) non-hummock poorly vegetated areas with *Dryas octopetala* L. and various herbs dominant or (3) hummock tundra with graminoid dominance (*Eriophorum vaginatum*) and creeping shrubs (*Salix* spp., *Betula nana*); (4) high dense *Pinus pumila* shrub associations; and (5) *Larix cajanderi* tree stands with different degrees of openness and different understorey compositions.

The predictions of the 40 new sample plots into RDA-space assigned two sample plots to the class "larch closed-canopy forest", 17 sample plots to "forest tundra and shrub tundra", 13 sample plots to "graminoid tundra", and seven sample plots to "prostrate herb tundra and barren" (Fig. 3). In situ AGB for each investigated 2018 expedition vegetation sample plot are published in Shevtsova et al. (2020c).

In the larch closed-canopy forest *L. cajanderi* makes the highest contribution to AGB (92% or 10.20 kg m$^{-2}$ (IQR=5.09 kg m$^{-2}$) on average of the total of 11.04 kg m$^{-2}$ (IQR=4.98 kg m$^{-2}$)). Other major vegetation groups are mosses and lichens (4%; 0.43 kg m$^{-2}$ (IQR=0.004 kg m$^{-2}$)) and low and dwarf shrubs (4%; 0.41 kg m$^{-2}$ (IQR=0.10 kg m$^{-2}$)), among them *Betula exilis* (0.21 kg m$^{-2}$ (IQR=0.017 kg m$^{-2}$)), *Ledum palustre* L. (0.10 kg m$^{-2}$ (IQR=0.019 kg/m$^2$)), *Vaccinium vitis-idaea* L. (0.08 kg m$^{-2}$ (IQR=0.061 kg m$^{-2}$)), *Salix* spp. (0.006 kg m$^{-2}$ (IQR=0.004 kg m$^{-2}$)), *Empetrum nigrum* L. (0.006 kg m$^{-2}$ (IQR=0.006 kg m$^{-2}$)), and *V. uliginosum* L. (0.003 kg m$^{-2}$ (IQR=0.003 kg m$^{-2}$)).

In the forest tundra and shrub tundra, 60% of the average sample plot AGB (1.44 kg m$^{-2}$ (IQR=2.40 kg m$^{-2}$)) is *Larix cajanderi* which accounts for 0.86 kg m$^{-2}$ (IQR=1.45 kg m$^{-2}$), followed by mosses and lichens (28%; 0.40 kg m$^{-2}$ (IQR=0.19 kg m$^{-2}$)). Low and dwarf shrubs are 10% (0.14 kg m$^{-2}$ (IQR=0.27 kg m$^{-2}$) of total sample plot AGB, among them *Betula nana* (0.05 kg m$^{-2}$ (IQR=0.09 kg/m$^2$)), *V. vitis-idaea* (0.04 kg m$^{-2}$ (IQR=0.06 kg m$^{-2}$)), *Ledum palustre* (0.03 kg m$^{-2}$ (IQR=0.05 kg m$^{-2}$)), *V. uliginosum* (0.02 kg m$^{-2}$ (IQR=0.06 kg m$^{-2}$)), *Salix* spp. (0.003 kg m$^{-2}$ (IQR=0.118 kg m$^{-2}$)) and *E. nigrum* (0.001 kg m$^{-2}$ (IQR=0.010 kg m$^{-2}$)). The remaining 2% (0.03 kg m$^{-2}$ (IQR=0.01 kg m$^{-2}$)) are mostly graminoids or other herbs.





In the graminoid tundra, 56% (0.25 kg m$^{-2}$ (IQR=0.32 kg m$^{-2}$)) of the average sample plot AGB (0.36 kg
m$^{-2}$ (IQR=1.49 kg m$^{-2}$)) are mosses and lichens, 20% (0.07 kg m$^{-2}$ (IQR=0.98 kg m$^{-2}$)) are low and dwarf
shrubs, and the remaining 10% (0.04 kg m$^{-2}$ (IQR=0.17 kg m$^{-2}$)) are other plants (grasses and forbs). Low
and dwarf-shrub contributors are *B. nana* (0.02 kg m$^{-2}$ (IQR=0.04 kg m$^{-2}$)), *L. palustre* (0.018 kg m$^{-2}$
(IQR=0.067 kg m$^{-2}$)), *Salix* spp. (0.019 kg m$^{-2}$ (IQR=0.03 kg m$^{-2}$)), *V. vitis-idaea* (0.013 kg m$^{-2}$
(IQR=0.019 kg m$^{-2}$)), and *V. uliginosum* (0.008 kg m$^{-2}$ (IQR=0.024 kg m$^{-2}$)).

The average (median) sample plot AGB of the prostrate herb tundra and barren areas is 0.11 kg m$^{-2}$
(IQR=0.25 kg m$^{-2}$) of which 82% is dwarf-shrub biomass with a dominance of *Dryas octopetala* (0.07 kg
m$^{-2}$ (IQR=0.08 kg m$^{-2}$)) and minor contributions of *V. uliginosum* (0.006 kg m$^{-2}$ (IQR=0.014 kg m$^{-2}$)), *V.
vitis-idaea* (0.005 kg m$^{-2}$ (IQR=0.005 kg m$^{-2}$)), *L. palustre* (0.002 kg m$^{-2}$ (IQR=0.008 kg m$^{-2}$)), and *Salix*
spp. (0.001 kg m$^{-2}$ (IQR=0.002 kg m$^{-2}$)). Moss and lichens account for 10% or 0.11 kg m$^{-2}$ (IQR=0.32 kg
m$^{-2}$) of the average sample plot AGB. The other 8% (0.08 kg m$^{-2}$ (IQR=0.08 kg m$^{-2}$)) of AGB is biomass
of different herbs.

Additionally, we analysed individual partial AGB of four taxa: *Larix cajanderi*, *Alnus fruticosa*, *Pinus
pumila*, and non-creeping *Salix* spp. (Fig. 6). *Pinus pumila* had a very wide range of needle to wood mass
ratios, including a ratio indicating a higher weight of needle biomass compared to wood biomass from an
individual shrub. For all other investigated species this is not the case. In contrast, deciduous-needled
larch has the lowest weight ratio of needles to wood when compared to *P. pumila*, *Salix* spp., and *A.
fruticosa*. In the different areas of investigation, we observe generally higher leaf (needle) to wood mass
ratios in the tundra–taiga area (16-KP-01) than in the northern taiga (18-BIL-01, 18-BIL-02).

### 3.2 Upscaling above-ground biomass using GAM

In the GAM, the RDA scores are explanatory variables and total AGB is the dependent variable. The first
two RDA axes explain 87% of the variance in the AGB data (Table 2). Both variables (parametric
coefficient RDA1 and the smooth term $s$(RDA1, RDA2)) are highly significant in the model.

We plotted fitted values against residuals for the GAM model to visualise residual standard deviations
(SD) for every sample plot used in the modelling (Fig. 7). There is some slight heteroscedasticity and the
SD increases with an increase of absolute AGB values. The RMSE of the model is 1.08 kg.





Based on the most recent Landsat data acquisitions, the maximum total AGB estimated within our study area is found in the northern taiga in the larch closed-canopy forests (20–24 kg m$^{-2}$, 16-KP-02, Fig. 8). In the southern tundra–taiga transition (16-KP-03) maximum AGB reached 12 kg m$^{-2}$ at places in a river valley that are covered by azonal dense forests. In the northern tundra–taiga (16-KP-01) maximum AGB is 4–6 kg m$^{-2}$ in the forest tundra and shrub tundra. In the tundra (16-KP-04) it is 3–4 kg m$^{-2}$ on the slopes of rivers' valleys.

### 3.3 Change of above-ground biomass between 2000 and 2017 in the four focus areas

The compiled change-maps of recent (20016/2017) versus 15–16 years earlier (2000/2001/2002) show the rates and spatial patterns of AGB change in the four focus areas (Fig. 9).

**Tundra area 16-KP-04, 2002–2017**: AGB of prostrate herb tundra vegetation has not changed within the investigated period (0 kg m$^{-2}$: IQR=0.12 kg m$^{-2}$ in 2002, IQR=0 kg m$^{-2}$ in 2017), AGB of graminoid tundra vegetation has slightly decreased (0.69 kg m$^{-2}$: IQR=0.83 kg m$^{-2}$ in 2002, 0.58 kg m$^{-2}$: IQR=0.99 kg m$^{-2}$ in 2017). A change in land-cover class from graminoid tundra to forest tundra and shrub tundra between 2002 and 2017 resulted in AGB increase from 1.42 kg m$^{-2}$ (IQR=0.49 kg m$^{-2}$) to 1.71 kg m$^{-2}$ (IQR=0.44 kg m$^{-2}$), whereas a change from prostrate herb tundra to graminoid tundra resulted in AGB decrease from 0.48 kg m$^{-2}$ (IQR=0.87 kg m$^{-2}$) to 0 kg m$^{-2}$ (IQR=0.23 kg m$^{-2}$).

**Northern tundra–taiga area 16-KP-01, 2001–2016**: AGB of prostrate herb tundra vegetation stayed stable at 0 kg m$^{-2}$ (IQR=0.29 kg m$^{-2}$ in 2001, IQR=0.34 kg m$^{-2}$ in 2016) on average, while the graminoid tundra AGB increased from 0.65 kg m$^{-2}$ (IQR=1.04 kg m$^{-2}$) to 1.40 kg m$^{-2}$ (IQR=0.48 kg m$^{-2}$) and the forest tundra and shrub tundra AGB has not changed (1.73 kg m$^{-2}$: IQR=0.50 kg m$^{-2}$ in 2001, 1.70 kg m$^{-2}$: IQR=0.32 kg m$^{-2}$ in 2016). A change in land-cover class from prostrate herb tundra into graminoid tundra resulted in AGB increase from 0 kg m$^{-2}$ (IQR=0.24 kg m$^{-2}$) in 2001 to 0.34 kg m$^{-2}$ (IQR=0.67 kg m$^{-2}$) in 2016, as did a change from graminoid tundra to forest tundra and shrub tundra from 1.27 kg m$^{-2}$ (IQR=0.53 kg m$^{-2}$) in 2001 to 1.69 kg m$^{-2}$ (IQR=0.29 kg m$^{-2}$) in 2016.

**Southern tundra–taiga area 16-KP-03, 2001–2016**: AGB of prostrate herb tundra vegetation has not changed and stayed at 0 kg m$^{-2}$ (IQR=0.50 kg m$^{-2}$ in 2001, IQR=0.31 kg m$^{-2}$ in 2016) on average, while graminoid tundra AGB increased from 1.00 kg m$^{-2}$ (IQR=0.91 kg m$^{-2}$) in 2001 to 1.50 kg m$^{-2}$ (IQR=0.57





kg m$^{-2}$) in 2016. The forest tundra and shrub tundra AGB has only slightly changed (2.00 kg m$^{-2}$: IQR=0.99 kg m$^{-2}$ in 2001, 2.10 kg m$^{-2}$: IQR=0.79 kg m$^{-2}$ in 2016). A change in land-cover class from

prostrate herb tundra to graminoid tundra resulted in AGB increase from 0.46 kg m$^{-2}$ (IQR=0.82 kg m$^{-2}$) in 2001 to 0.88 kg m$^{-2}$ (IQR=1.03 kg m$^{-2}$) in 2016 and a change from graminoid tundra to forest tundra and shrub tundra resulted in AGB increase from 1.43 kg m$^{-2}$ (IQR=0.48 kg m$^{-2}$) in 2001 to 2.02 kg m$^{-2}$ (IQR=0.66 kg m$^{-2}$) in 2016. A major AGB change is associated with forest tundra and shrub tundra becoming larch closed-canopy forest resulting in AGB increase from 3.02 kg m$^{-2}$ (IQR=1.29 kg m$^{-2}$) in

2001 to 7.29 kg m$^{-2}$ (IQR=2.53 kg m$^{-2}$) in 2016.

**Northern taiga area 16-KP-02, 2000–2016**: AGB of prostrate herb tundra vegetation increased from 0 kg m$^{-2}$ (IQR=0.09 kg m$^{-2}$) to 0.60 kg m$^{-2}$ (IQR=2.60 kg m$^{-2}$) ; graminoid tundra AGB increased from 1.30 kg m$^{-2}$ (IQR=0.82 kg m$^{-2}$) to 1.90 kg m$^{-2}$ (IQR=0.69 kg m$^{-2}$); forest tundra and shrub tundra AGB slightly increased from 2.70 kg m$^{-2}$ (IQR=1.33 kg m$^{-2}$) to 3.10 kg m$^{-2}$ (IQR=1.09 kg m$^{-2}$); and larch closed-canopy

forest AGB increased from 7.00 kg m$^{-2}$ (IQR=2.49 kg m$^{-2}$) to 7.50 kg m$^{-2}$ (IQR=4.65 kg m$^{-2}$) within the time studied. A change in land-cover class from prostrate herb tundra largely into graminoid tundra resulted in AGB increase from 0 kg m$^{-2}$ (IQR=0.08 kg m$^{-2}$) in 2000 to 1.45 kg m$^{-2}$ (IQR=0.93 kg m$^{-2}$) in 2016 and a change from graminoid tundra to forest tundra and shrub tundra resulted in AGB increase from 1.44 kg m$^{-2}$ (IQR=0.61 kg m$^{-2}$) in 2000 to 2.78 kg m$^{-2}$ (IQR=0.96 kg m$^{-2}$) in 2016. Some areas

classed as forest tundra and shrub tundra became larch closed-canopy, which resulted in AGB increase from 3.25 kg m$^{-2}$ (IQR=1.49 kg m$^{-2}$) in 2000 to 7.20 kg m$^{-2}$ (IQR=4.12 kg m$^{-2}$) in 2016.

AGB of land-cover classes that did not change within the investigated period tend to have higher values moving from the tundra to northern taiga (Fig. 10).

We find an increase in AGB for those areas where land-cover class has changed (Table 3). The highest

changes in the paired years occurred in the southern tundra–taiga (16-KP-03; +4.30 kg m$^{-2}$) and the northern taiga (16-KP-02: +4.09 kg m$^{-2}$) associated with a change in land-cover class from forest tundra and shrub tundra to larch closed-canopy forest. The lowest AGB change rates are associated with a change in land-cover class from graminoid tundra to forest tundra and shrub tundra in the northern taiga (16-KP-02) and southern tundra–taiga (16-KP-03). In general, total AGB in the tundra focus area has not changed

over the time studied (0 kg m$^{-2}$, IQR=0.2 kg m$^{-2}$), while in the northern tundra-taiga it has increased by





0.69 kg m$^{-2}$ (IQR=0.69 kg m$^{-2}$) and by 0.44 kg m$^{-2}$ (IQR=0.91 kg m$^{-2}$) in the southern tundra-taiga. In the northern taiga total AGB has increased much more than in the other focus areas by 1.3 kg m$^{-2}$ (IQR=1.4 kg m$^{-2}$).

## 4 Discussion

### 4.1 Recent state of above-ground biomass at the field sites

We estimated total and partial dry AGB for the 2018 expedition sample plots, which cover a wide range of vegetation associations (Shevtsova et al., 2020c; Shevtsova et al., 2020d). From these field biomass samples, AGB estimates range from 0 to 15 kg m$^{-2}$ and, as expected, reflect a gradient of land-cover classes from the least vegetated prostrate herb tundra and barren areas to the larch closed-canopy forests.

As in other subarctic and arctic vegetation studies the taxa found in our study region can be grouped into similar PFTs for a convenient comparison. Thus, deciduous shrubs are largely represented by *Betula nana*, *Vaccinium uliginosum* and *Salix* sp., which are typical circumpolar subarctic species (Grigoryev, 1946) and are widely found, for example in the tundra in Alaska near Toolik Lake (Shaver and Chapin, 1991). In graminoid tundra, which, by its characteristics, is comparable to tussock tundra in Alaska, deciduous

shrubs contribute 33% to the total AGB (tundra, median=0.09 kg m$^{-2}$, IQR=0.05 kg m$^{-2}$) or 9% (tundra-taiga, median=0.07 kg m$^{-2}$, IQR=0.05 kg m$^{-2}$), which is similar to deciduous shrub AGB of Alaskan tussock tundra (0.09±0.02 kg m$^{-2}$). However, in Alaska, deciduous shrub contribution to the total AGB is 16%, which is lower than the central Chukotka graminoid tundra, but higher than the graminoid tundra in the central Chukotkan tundra-taiga. Evergreen shrub taxa are also similar in our study region to those near

Toolik Lake, Alaska being mainly represented by *Ledum palustre*, *Vaccinium vitis-idaea*, *Dryas octopetala*, and *Empetrum nigrum* with *Pinus pumila* in our study region in contrast to Alaska. Evergreen shrubs generally have a lower AGB in the graminoid tundra of our study region (tundra, median=0.08, IQR=0.11; tundra-taiga, median=0.03, IQR=0.10) than in the tussock tundra of Alaska (0.17±0.02 kg m$^{-2}$), but the percentage of this PFT is slightly higher (31%) in central Chukotka than in Alaska (24%).

In the graminoid tundra of the central Chukotka tundra-taiga, AGB of evergreen shrubs is poorly represented (4%). Graminoids in our region were not separately sampled but are included as "other".





However, especially in graminoid tundra, the "other" class mostly consists of graminoids and other taxa inclusions are rare, so it can be a good approximation of graminoid AGB. The main taxa here, as in Alaska, are *Carex* sp. and *Eriophorum vaginatum*. Compared to the tussock tundra in the Toolik Lake

vicinity in Alaska, graminoid tundra of both tundra and tundra-taiga areas in central Chukotka has much less graminoid AGB. For the tundra area it is 9% of total AGB (median=0.02 kg m$^{-2}$, IQR=0.11 kg m$^{-2}$) and in the tundra-taiga it is 5% (median=0.04 kg m$^{-2}$, IQR=0.14 kg m$^{-2}$), whereas in Alaskan tussocks it is 16% of the total AGB (0.11±0.02 kg m$^{-2}$). All vascular plant AGB is similar for all compared areas of graminoid/tussock tundra. Graminoid tundra AGB contribution in the tundra area in central Chukotka is

0.25 kg m$^{-2}$ (median, IQR=0.04 kg m$^{-2}$) and in the tundra-taiga area it is 0.34 kg m$^{-2}$ (median, IQR= 2.46 kg m$^{-2}$, the high IQR is caused by *P. pumila* contributions at two sites). This compares to AGB of 0.37±0.03 kg m$^{-2}$ in the tussock tundra of Alaska. The contribution of vascular plants versus non-vascular plants is much higher in the graminoid tundra of the Chukotka tundra area (96%) than in Alaska (53%), whereas for the graminoid tundra of the Chukotka tundra-taiga ecotone their contribution is similar to

Alaska (42%). Total AGB of graminoid tundra in central Chukotka is strongly different between tundra (median=0.26 kg m$^{-2}$) and tundra-taiga (median=0.81 kg m$^{-2}$), with the latter being similar to total AGB of the Alaskan tussock tundra (0.71 kg m$^{-2}$), while the former is similar to total AGB of open areas and open north-boreal fen in northern Finland (0.30 kg m$^{-2}$, Räsanen et al, 2018). However, major taxa such as *Betula nana*, *Salix* sp. and graminoids have different contributions in these investigated areas. The

tundra area in central Chukotka (only graminoid tundra class) has higher AGB from *B. nana* (median=0.07 kg m$^{-2}$, IQR= 0.03 kg m$^{-2}$) and *Salix* sp. (median=0.01 kg m$^{-2}$, IQR=0.009 kg m$^{-2}$) than these taxa in northern Finland (0.02±0.05 kg m$^{-2}$ and 0.0005±0.008 kg m$^{-2}$, respectively), but similar AGB of graminoids (0.02 kg m$^{-2}$, IQR=0.11 kg m$^{-2}$ versus 0.03±0.011 kg m$^{-2}$).

The highest contribution to partial AGB in central Chukotka is from Cajander larch (*Larix cajanderi*), the

only tree species present in the study region. Despite many studies using complex allometric equations, mostly including tree height and stem diameter (e.g. Dong at al., 2020; Alexander et al., 2012; Bjarnadottir et al., 2007) to estimate AGB of an individual tree, we used only tree height because stem diameter measurements (stem perimeter) were not available for all trees. However, where measurements of tree stem diameters were available, these are shown to be highly correlated with height, which makes





it rational to use only height to estimate tree AGB to avoid multicollinearity in the model. Other parameters (crown height, crown width) were also measured on a subset of trees and proved to be insignificant predictors. Thus, using estimated tree height we provide coherent AGB estimation models by accounting for living state (live or dead) and ecological zone (tundra–taiga, northern taiga). We also estimated leaf and wood biomass separately and summed them up in the data processing procedure

(Appendix A). These established allometric equations can be applied at a broad scale in central Chukotka to a range of tree heights (up to 20 m), as covered by our study.

## 4.2 Recent state of above-ground biomass upscaled for central Chukotka

The AGB of the studied focus areas of central Chukotka varies along a gradient from <0.5 kg m$^{-2}$ in the sparsely vegetated areas of the tundra to 25 kg m$^{-2}$ in the dense larch forests of the northern taiga. When

comparing areas in the circumpolar region with a similar vegetation to our study region it can be seen that graminoid tundra in central Chukotka generally has less AGB than tussock tundra in Alaska (Toolik research station, Shaver and Chapin, 1991), whereas forest tundra in central Chukotka has more larch AGB than the Kolyma region (Berner et al, 2018).

Circumpolar remote sensing-based estimations such as in Santoro and Cartus (2019) and Raynolds et al.

(2011) have lower spatial resolution and less precise AGB estimates for central Chukotka than our mapped AGB estimates. The most recent (2017) European Space Agency (ESA) global AGB map (Santoro and Cartus, 2019) shows generally lower AGB estimates for non-mountainous regions of central Chukotka than our AGB estimates: shrublands in tundra with an AGB of 1.5–4 kg m$^{-2}$ (our estimations) only range from 0.3 to 0.6 kg m$^{-2}$ in the ESA AGB product; our AGB estimates for forest tundra in the

tundra-taiga ecotone range from 2.5 to 3 kg m$^{-2}$ but are 0.07–0.16 kg m$^{-2}$ in the ESA AGB product; for graminoid tundra in the tundra-taiga ecotone our AGB estimates are 0.7–3 kg m$^{-2}$ while ESA AGB is 0.1–0.8 kg m$^{-2}$; and our larch closed-canopy forests AGB estimates are 22–24 kg m$^{-2}$ versus 2.8–4 kg m$^{-2}$ in the ESA. In contrast, mountainous regions show unrealistically high AGB values in the ESA AGB product that are most likely due to topographical artefacts in the Synthetic Aperture Radar (SAR)

processing of the ESA AGB product (see also Santoro and Cartus, 2019). However, other spatial distribution patterns of AGB, especially in the tundra–taiga areas (16-KP-01, 16-KP-03) are very similar





to our AGB results. The dissimilarities in the AGB magnitudes can be explained by the different remote-sensing methods: the ESA AGB product was derived from SAR remote sensing while our AGB estimates are based on optical Landsat data. SAR-based biomass estimation is sensitive to vegetation structure and

can only derive higher vegetation layers. Therefore, ESA AGB can only represent a 'living tree AGB', while our AGB estimates include other plant groups (lower shrubs, ground vegetation, mosses and lichens) of central Chukotka and are thus more suitable for the investigated area.

Two of our focus areas overlap with the circumpolar above-ground phytomass map of peak-summer season (Raynolds et al., 2011) and a comparison reveals that AGB estimates for the tundra–taiga area

(16-KP-01) are similar to each other: 0.65 kg m$^{-2}$ (IQR=1.1 kg m$^{-2}$) in 2001 and 1.5 kg m$^{-2}$ (IQR=0.46 kg m$^{-2}$) in 2016 (our estimates) versus 0.61–0.97 kg m$^{-2}$ in 2010. However, for the second area, 16-KP-04, our average AGB estimate is lower during the whole investigation period at 0 kg m$^{-2}$ (IQR=0.7 kg m$^{-2}$) in 2002 and 0 kg m$^{-2}$ (IQR=0.37 kg m$^{-2}$) in 2017 versus 0.61–0.97 kg m$^{-2}$ in 2010 as estimated by Raynolds et al. (2011).

Further comparison with AGB of similar vegetation types in Alaska (Toolik research station, Shaver and Chapin, 1991) shows that tussock tundra has higher AGB in Alaska (0.71 kg m$^{-2}$) than graminoid tundra in central Chukotka (0.36 kg m$^{-2}$), despite having a similar composition that includes tussocks and is also dominated by *Eriophorum vaginatum*. This may be because the AGB of graminoids and forbs in Alaska (0.12 kg m$^{-2}$) is higher than in central Chukotka (0.04 kg m$^{-2}$) as is the AGB of dwarf shrubs (0.26 kg m$^{-2}$

versus 0.07 kg m$^{-2}$). The "prostrate herb tundra and barren areas" land-cover class in central Chukotka has a similar composition to heath communities in Alaska with evergreen dwarf shrubs and extensive exposed ground. Prostrate herb tundra AGB of central Chukotka is lower (0.11 kg m$^{-2}$) compared to Alaska (0.32 kg m$^{-2}$), having more lichen and dwarf-shrub biomass. Forest tundra and shrub tundra in central Chukotka is challenging to compare to Alaskan communities, but generally, average AGB in this

land-cover class is slightly lower (1.33 kg m$^{-2}$) than AGB of even shrub-only communities in Alaska (1.39 kg m$^{-2}$), which are formed of tall deciduous shrubs such as *Salix* spp. growing on river bars and well-drained floodplains. In contrast to Alaska, forest tundra and shrub tundra in central Chukotka includes mostly dwarf or sparse low shrubs, as well as some tall shrubs and open larch tree stands, and are found on more diverse landscape features than river bars. In addition, the AGB of the central Chukotka tundra



and also, partly, the northern tundra–taiga is generally comparable to the AGB of the North Slope of Alaska, which ranges from 0 to 4 kg m$^{-2}$ (Berner et al, 2018).

Comparing our AGB estimates of *Larix cajanderi* to those in the area around the Kolyma River (western Chukotka; Berner et al., 2012) – a close match to our study region by vegetation composition and partly environmental settings – reveals similarities in the spatial patterns of AGB distribution. Highest AGB

tends to occur on protected mountain valley slopes in both investigated regions. AGB of *Larix cajanderi* open forests in the Kolyma river area ranges, on average, from 0.5 to 5 kg m$^{-2}$, reaching the maximum of 6.7 kg m$^{-2}$, which is comparable with our forest tundra and shrub tundra AGB assuming a 57% representation of *Larix cajanderi* in this land-cover class.

Many factors can influence the AGB estimates such as the number of reference samples, prediction

method, remote sensing sensor type (optical, radar), as well as spatial and temporal resolution of the satellite imagery and products (Fassnacht et al., 2014). Overall, a comparison with global and circumpolar AGB estimates highlights great improvements in the accuracy of the estimates and a better way to resolve a more landscape-related spatial pattern of our AGB estimates for the study region.

## 4.3 Change in above-ground biomass within the investigated 15–16 years in central Chukotka

We derived total AGB changes in the central Chukotka from Landsat satellite data spanning 15–16 years and found the greatest change in the dense forests of the northern taiga (16-KP-02). In the northern tundra–taiga area (16-KP-01), AGB increased from 2001 to 2016 by 0.046 kg m$^{-2}$ yr$^{-1}$ (IQR= 0.046 kg m$^{-2}$yr$^{-1}$), which is much faster than the rate estimated by Epstein et al. (2012) for 1982 to 2010 (0.004–0.015 kg m$^{-2}$ yr$^{-1}$). Further, we estimated AGB change from 2002 to 2017 in the tundra focus area (16-KP-04) as

being close to 0 kg m$^{-2}$yr$^{-1}$ (IQR= 0.013 kg m$^{-2}$yr$^{-1}$) on average, which is lower than estimations from 1982 to 2010 given in the circumpolar above-ground phytomass map for the Russian Far East (Walker and Raynolds, 2018). Our results of tundra AGB change being close to zero are similar to experiments with modelling extreme temperature increases in Alaskan tundra (Hobbie and Chapin, 1998). In their study, Hobbie and Chapin (1998) conclude that, in tundra, plant biomass accumulation depends on

nutrient availability and AGB will only increase if mineralisation of soil organic nutrients is stimulated together with climate warming. Given differences in soil development between the focus areas of tundra,



tundra–taiga and northern taiga, their conclusion may also apply to our results. In general, the comparison with circumpolar estimated AGB changes from 1982 to 2010 (Walker and Raynolds, 2018) shows that changes in AGB in our focus areas of central Chukotka between 2000 and 2017 were much faster,

probably because of the stronger warming in the first decades of the 21$^{st}$ century in these regions.

Our estimates of AGB change within our land-cover classes show that AGB change does not necessarily lead to a change in land-cover class. We assume that changes for different regions within the same stable land-cover classes could be associated with population size change, but also, likely, with changes in the plant's parameters (height, crown density etc.). This could explain why the change in AGB estimated for

the graminoid tundra in the northern taiga (16-KP-02) is greater than for the tundra (16-KP-04, Fig.10).

**Conclusions**

We successfully used field-based AGB data and Landsat satellite data in statistical modelling to map recent (2016/2017) and historical (2000/2001/2002) states of AGB in four focus areas along a tundra–taiga gradient in central Chukotka. The total AGB values consist of major taxon-specific (and other)

estimates that allow us to achieve a more detailed picture of AGB change and to reveal changes in major species contributions from areas with diverse ecology. In addition, we were able to analyse changes in AGB together with changes in land-cover classes.

AGB of the investigated areas in the field ranged from 0 to 16 kg m$^{-2}$. Taxa making the most contribution to AGB in our study region include Cajander larch (*Larix cajanderi*) in forest stands, and dwarf birch,

dwarf willows, heathers, *Dryas octopetala* (only in prostrate herb tundra and barren areas), mosses, and lichens in tundra areas. Forested sites generally had higher AGB (2.38 kg m$^{-2}$, IQR= 3.06 kg m$^{-2}$) than open tundra (hummocks with dwarf or low shrubs 0.65 kg m$^{-2}$, IQR= 0.76 kg m$^{-2}$; prostrate tundra 0.32 kg m$^{-2}$, IQR=0.22 kg m$^{-2}$). Tall *Pinus pumila* shrub communities have the highest total AGB (5.57 kg m$^{-2}$, IQR=1.14 kg m$^{-2}$), but are rare at the landscape level and are azonal. Thus, an expansion of forest would

make the strongest change to total ABG, but it is still unclear how fast taiga could colonise tundra areas in the upcoming decades. Nevertheless, taxon-specific estimations allow us to separate tree biomass from other vegetation forms, expanding the usefulness of our study to treeline migration assessment and forest management in the study region.





Estimation of recent AGB (2016/2017) in our four focus areas found the highest AGB (24 kg m$^{-2}$) in the
larch closed-canopy forests of the southern tundra–taiga and northern taiga. The lowest AGB occurred in
the prostate herb tundra and barren land-cover class and largely in the tundra on a landscape scale. On
average, above-ground vegetation of the closed-canopy forest class has an AGB of 8.9 kg m$^{-2}$ (IQR= 6.4
kg m$^{-2}$), the forest tundra and shrub tundra class 3.3 kg m$^{-2}$ (IQR= 1.2 kg m$^{-2}$), the graminoid tundra class
1.4 kg m$^{-2}$ (IQR= 0.53 kg m$^{-2}$), and the prostrate herb tundra and barren areas class close to 0 kg m$^{-2}$
(IQR= 0 kg m$^{-2}$; for non-barren areas 0.4 kg m$^{-2}$, IQR=0.52 kg m$^{-2}$). A comparison with other available
estimations of AGB for central Chukotka revealed that other studies considerably overestimate
mountainous prostate herb tundra and barren areas and underestimate tundra-taiga and northern taiga
areas. Our satellite-derived estimations match the magnitude of the ground data and show greater detail
in the spatial phytomass distribution for the study region.

We found that the greatest AGB changes occurred in the northern taiga, particularly in the larch closed-
canopy forest class (+4.09 kg m$^{-2}$), which also has the highest AGB and most favourable environment for
the expansion of *Larix cajanderi* which contributes highly (92% on average) to AGB. The less favourable
environments in the tundra–taiga and tundra would need more time to adapt to recent climate changes.
We found changes in AGB that are not only associated with changes in land-cover classes, but also within
areas with no changes in land-cover class. This could indicate either that vegetation composition changes
are not yet prominent enough to trigger a change in land-cover class, or that there has been a change in
plant properties (height, crown diameter, leaf size etc.) within the investigated period.

Overall, our mapped AGB of recent and historical times in central Chukotka are of value in helping to
understand regional ecosystem dynamics as well as circumpolar processes, especially in the light of recent
climate changes. The specific parameterisation of plant biomass from central Chukotka make our AGB
maps most suitable for the region and more precise in terms of spatial resolution than global and
circumpolar estimations of AGB. Future uses of our AGB state and change maps could include modelling
of carbon stocks and investigating habitat changes in the area. Knowing the recent and historical AGB
distribution and the contributing taxa is useful for modelling studies that aim to project future AGB
changes, as well as for policy-making, particularly in relation to mitigation of climate-change impacts
and conservation.





### Appendix A. Sampling and above-ground biomass (AGB) calculation protocol for field data

Here we present the step-by-step protocol for harvesting and calculating ground-layer AGB for a 30 x 30 m sample plot in kg m$^{-2}$:

1)      Fresh biomass harvested and weighed (sample of a particular taxon from a 0.25 m$^2$ plot): $GFW$

2)      Fresh biomass subsample from the $GFW$ sample: $subFW$ (g/0.25 m$^2$)

3)      Dry biomass from the subsample: $subGDW$ (g/0.25 m$^2$)

4)      Dry weight from the sample (g/0.25 m$^2$):

$$GDW = \frac{GFW * subGDW}{subGFW} ,$$  (A1)

for moss samples

$$GDW = \frac{GFW * subGDW}{0.04 subGFW} .$$  (A2)

5)      Dry weight of all samples per subplot *sub B* (as in Fig. 2, kg m$^{-2}$)

$$GDWsubplotb = 0.004 \sum_1^k GDW ,$$  (A3)

$k$ is number of taxa sampled on the subplot B

6)      Total dry weight for the whole 30 x 30 m plot (kg 30 m$^2$):

$$GDWplot = 9a * GDWsubplotb1 + 9b * GDWsubplotb2,$$  (A4)

$a$ and $b$ are proportions of vegetation represented by subplot B1 and B2 (estimated subjectively during field data inventory) on the 30 x 30 m plot, respectively.

7)      Average total dry weight (kg m$^{-2}$):


$$GDWavg = \frac{GDWplotplot}{900},$$  (A5)

**Calculation for *Pinus pumila* shrub AGB.** We sampled three (small, medium, big) individual pine plants on each 30 x 30 m sample plot that contained the species. With the following steps we calculated the AGB for each individual plant:

1)      Woody AGB of all small living branches (g):

$$DWSmBrsB\ (S, M\ or\ B) = \frac{nSBr(FWSmBrB * subFWSmBrB)}{subDWSmBrB},$$  (A6)

where $subDWSmBrB$ is dry weight of subsample of a small branch wood; $S$, $M$ or $B$ are size of an individual plant; $nSBr$ is the number of small branches, $FWSmBrB$ or $subFWSmBrB$ is the fresh weight of a whole sample or subsample of a small branch wood, respectively.

2)      Needle AGB of all small living branches (g):

$$DWSmLsB\ (S, M\ or\ B) = \frac{nSBr(FWSmLB * subFWSmLB)}{subDWSmLB},$$  (A7)

where $subDWSmLB$ *is* dry weight of subsample of a small branch needles, $FWSmLB$ or $subFWSmLB$ is the fresh weight of a whole sample or subsample of a small branch needles, respectively.





3)      Woody AGB of all big living branches (g):

$$DWBiBrsB \ (S, M \ or \ B) \ = \frac{nBiBr(FWBiBrB \ *subFWBiBrB)}{subDWBiBrB},$$                                         (A8)

where  $subDWBiBrB$ is dry weight of subsample of a big branch wood, $nBiBr$ is the number of big branches, $FWBiBrB$ or $subFWBiBrB$ is the fresh weight of a whole sample or subsample of a big branch wood, respectively.

4)   Woody AGB of all dead branches (g):

$$DWdBrsB \ (S, M \ or \ B) \ = \frac{ndBr(FWdBrB \ *subFWdBrB)}{subDWdBrB},$$                                            (A9)

 where  $subDWdBrB$ is dry weight of subsample of a big branch wood, $ndBr$ is the number of dead branches, $FWdBrB$ or

$subFWdBrB$ is the fresh weight of a whole sample or subsample of a dead branch wood, respectively.

5)      Average AGB of small living branch wood (across the three different-sized samples, g):

$$DWSmBrsBAv \ = \frac{DWSmBrsB \ (S) + DWSmBrsB \ (M) + DWSmBrsB \ (B)}{3}.$$                                         (A10)

6)      Average AGB of small living branch needles (g):

$$DWSmLsBAv \ = \frac{DWSmLsB \ (S) + DWSmLsB \ (M) + DWSmLsB \ (B)}{3}.$$                                            (A11)

7)      Average AGB of big living branch wood (g):

$$DWBiBrsBAv \ = \frac{DWBiBrsB \ (S) + DWBiBrsB \ (M) + DWBiBrsB \ (B)}{3}.$$                                          (A12)

8)   Average AGB of dead branch wood (g):

$$DWdBrsBAv \ = \frac{DWdBrsB \ (S) + DWdBrsB \ (M) + DWdBrsB \ (B)}{3}.$$                                              (A13)

9)      Average individual plant wood total AGB (including cones biomass, g):

$$AvWoodDW \ = \ DWSmBrsBAv \ + \ DWBiBrsBAv \ + \ DWdBrsBAv + nc * cB$$                                                ,

(A14)

where $AvWoodDW$ is the average dry weight for only the woody part of a plant, $nc$ is number of cones, and $cB$ is cones biomass.

10)      Average volume of a shrub crown (cm$^3$):

$$CrV \ = \frac{SH*SCr1* \ SCr2 + \ MH*MCr1*MCr2+ \ BH*BCr1*BCr2}{3},$$                                              (A15)

where $SH$, $MH$ and $BH$ is height of a small, medium and big plant respectively; $Cr1$ and $Cr2$ are two measurements of a diameter of a crown perpendicular directions.

11)      Average wood AGB of *Pinus pumila* (g m$^{-2}$):

$$DWAvWood \ = \ AvWoodDW \ *\frac{10000}{CrV},$$                                                                      (A16)

where $DWAvWood$ is the average woody mass of a plant per m$^2$.

12)      Average needle AGB of *Pinus pumila* (g m$^{-2}$):

$$DWAvLs \ = \ DWSmLsB \ *\frac{10000}{CrV},$$                                                                        (A17)

where $DWAvLs$ is the average needles' mass of a plant per m$^2$.



13)    Total average AGB of *Pinus pumila* shrub on a  30 x 30 m sample plot (kg m$^{-2}$):

$TDAGBPp = 0.1\,e(DWAvWood + DWAvLs)$,                                                    (A18)

*TDAGBPp* is the total average AGB of a plant on the 30 x 30 m sample plot, *e* is cover of *Pinus pumila* shrubs on the 30 x 30 m sample plot (%).

**Calculation for *Alnus fruticosa* and *Salix* sp. shrubs AGB.** We sampled three (small, medium, big) individuals as for *Pinus*

*pumila* at each plot if present. Calculations are similar as for pine, but include not only big and small branches, but also medium branches.

**Calculation for *Larix cajanderi* AGB.** *Larix cajanderi* trees were representatively subsampled at the following parts: living branches (small, medium, big), dead branches, needles from small branches, stem (ideally three tree discs at 0, 1.3, and 2.6 m

heights), and cones. Total AGB of an individual tree (g) from the field survey of 2018 expedition was calculated as following:

1)   $TDAGB = DBrLB + DTrB$,                                                                   (A19)

where *TDAGB* is total dry AGB of a tree, *DBrLB* is dry weight of biomass of branches and leaves, *DTrB* is dry weight of stem biomass.

2)   $DBrLB = nSBr * SmBrB + nSBr * SmLB + nMBr * MBrB +$

545                 $+ nBiBr * BiBrB + ndBR * dBrB + nc * cB$,                                 (A20)

where *nSBr* is the number of small branches, *SmBrB* is the small branch dry biomass,  *SmLB* is the small branch needles dry biomass, *nMBr* is the number of medium branches, *MBrB* is medium branch dry biomass, *nBiBr* is number of big branches, *BiBrB* is dry biomass of big branches, *ndBR* is number of dead branches, *dBrB* is dead branch biomass, *nc* is number of cones, and *cB* is cones biomass.

3)   $DTrB = V_{A-B} * TrDens_{A-B} + V_{B-C} * TrDens_{B-C} + V_C * TrDens_C$,             (A21)

where *V* is volume (  $_{A-B}$ is a base of a tree stem from 0 to 130 cm,   $_{B-C}$ is a middle part of a tree stem from 130 to 260

cm,   $_C$ is a top part of a tree stem from 260 to the top), *TrDens* is the wood density of a tree part (base, middle or top).

4)   $TrDens_{A-B} = \frac{TrDens_A + TrDens_B}{2}$,                                            (A22)

where $TrDens_A$ is the wood density of tree disc A and $TrDens_B$ is the wood density of tree disc B.

5)   $TrDens_{B-C} = \frac{TrDens_B + TrDens_C}{2}$,                                       (A23)

where $TrDens_C$ is the wood density of a tree disc C.

6)   $TrDens_A = \frac{V_{Adisc}}{B_{Adisc}} = \pi\,h_{Adisc}(\frac{DAdisc}{2})^2 - \pi\,h_{Adisc}(\frac{Dz}{2})^2 - Crl * Crw * h_{Adisc}$,   (A24)

where $V_{Adisc}$ is the volume of a tree disc sampled at 0 cm tree stem height, $B_{Adisc}$ is dry weight of a tree disc sampled at 0 cm tree stem height, $h_{Adisc}$ is height of a tree disc sampled at 0 cm tree stem height, $D_{Adisc}$ is diameter of a tree disc sampled at 0

cm tree stem height, $D_z$ is diameter of a circular hole in the central part of a disc (if present), and *Crl* and *Crw* are length and





average width of a crack in the tree disc, respectively (if present). $TrDens_B$ and $TrDens_C$ are calculated by analogy with $TrDens_A$.

7) Calculation of volume of a tree part (base, middle or top) varies depending on presence or absence of a central hole in the tree stem.

Scenario 1: A hole in the tree disc is absent $Dz = 0$:

$$V_{A-B} = \frac{130\pi}{3}\left(\left(\frac{D_A}{2}\right)^2 + \left(\frac{D_B}{2}\right)^2 + \left(\frac{D_A*D_B}{4}\right)\right),$$ (A25)

where $V_{A-B}$ is the volume of a tree stem part from 0 (A) to 130 cm (B), $D_A$ is diameter of disc A, and $D_B$ is diameter of disc B.

$$V_C = \frac{\pi(H-260)}{3} * \left(\frac{DC}{2}\right)^2,$$ (A26)

where $V_c$ is the volume of a top part of a tree stem from 260 cm to the full height of a tree ($H$) and $D_C$ is the diameter of disc

C.

Scenario 2: A hole in the tree disc is present only in disc A $Dz \neq 0$ (only A):

$$V_{A-B} = \frac{130\pi}{3}\left(\left(\frac{D_A}{2}\right)^2 + \left(\frac{D_B}{2}\right)^2 + \left(\frac{D_A*D_B}{4}\right)\right) - \frac{130\pi}{3}\left(\frac{Dz_A}{2}\right)^2,$$ (A27)

where $Dz_A$ is the diameter of a central circular hole in disc A.

$V_c$ – by analogy with Scenario 1.

Scenario 3: A hole in the tree disc is present in discs A and B $Dz \neq 0$ (A and B):

$$V_{A-B} = \frac{130\pi}{3}\left(\left(\frac{D_A}{2}\right)^2 + \left(\frac{D_B}{2}\right)^2 + \left(\frac{D_A*D_B}{4}\right)\right) - \frac{130\pi}{3}\left(\left(\frac{Dz_A}{2}\right)^2 + \left(\frac{Dz_B}{2}\right)^2 + \left(\frac{Dz_A*Dz_B}{4}\right)\right),$$ (A28)

where $Dz_B$ is the diameter of a central circular hole in disc B.

$V_c$ – by analogy with Scenario 1.

The next step in estimation of *Larix cajanderi* AGB was to estimate it for the 30x30 m sample plot, limited to tree height as a predictor. We differentiated between allometric equations to estimate partial individual larch AGB from trees from two ecological regions (tundra–taiga and northern taiga).

To assess the different models for different regions we used a Wilcoxon rank sum test on measurements of tree stem perimeters. It showed significant differences between basal perimeter and perimeter at 1.3 m height of trees from 16-KP-01 (tundra–taiga,

178 samples) and BIL-18 (northern taiga, 74 samples) (Fig. B1). In both cases, tree basal perimeter ($p=0.007453$) and tree perimeter at 1.3 m ($p=0.03014$) in the tundra–taiga is statistically greater than in northern taiga. Since individual trees are significantly different in the two regions, different AGB-prediction models are required for the tundra–taiga and northern taiga focus areas.





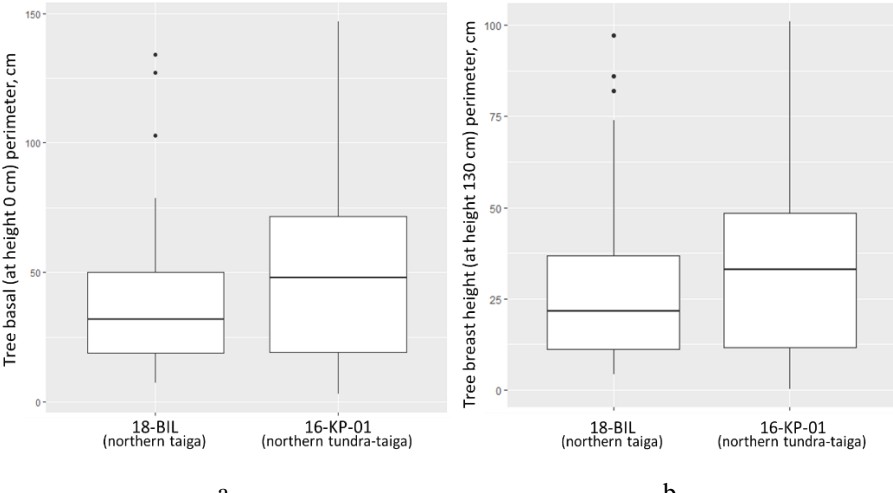

590          a          b

**Figure B1. Distribution of basal (a) and breast height (b) diameters values of trees from two focus areas: northern taiga (18-BIL) and northern tundra–taiga (16-KP-01). We also made separate models for living and dead trees as there are obvious differences in the wood densities and no needle material for dead trees. Total AGB of a tree was calculated from partial needle and wood biomass estimations.**


Below are the allometric equations that we established:

1)        Needle biomass of a living tree (area 16-KP-01, g):

$$AGBn\,(16-KP-01) \;=\; \frac{703.62}{1+e^{-\frac{H-579.5}{208.69}}}\,,$$ (A29)

where AGB is above ground biomass and $H$ is tree height in cm (Kruse et al., 2020).

2)        Needle biomass of a living tree (areas 18-BIL-01/18-BIL-02, g):

$$AGBn\,(18-BIL) \;=\; 12.176e^{0.0029H}$$ (A30)

3)        Needle biomass of a dead tree (both regions, g): $AGBnd \;=\; 0$ (A31)

4)        Wood biomass of a living tree (area 16-KP-01, g):

$$AGBwl\,(16-KP-01) \;==\; \frac{78713.63}{1+e^{-\frac{H-793.64}{73.91}}}$$ (A32)

5)        Wood biomass of a living tree (area 18-BIL, g):

$$AGBwl\,(18-BIL) \;=\; 170.69e^{0.0046H}$$ (A33)

6)        Wood biomass of a dead tree (both areas, g):

$$AGBwd \;=\; 203.3e^{0.0057H}$$ (A34)

*Larix cajanderi* AGB for a 30 x 30 m sample plot was calculated as following:

$LCAGB = \sum_1^k AGBn + \sum_1^k AGBw,$



where $k$ is number of trees on the 15 m radius sample plot, $AGBn$ is the needle biomass of a tree, and $AGBw$ is the woody biomass of a tree.

**Data availability statement**

The data that support the findings of this study are published in the PANGAEA® Data Repository for Earth and Environmental
Science. The following data sets are accessible via the stated links:

1. Kruse, S., Herzschuh, U., Schulte, L., Stuenzi, S. M., Brieger, F., Zakharov, E. S., Pestryakova, L. A.: Forest inventories on circular plots on the expedition Chukotka 2018, NE Russia, PANGAEA, doi.pangaea.de/10.1594/PANGAEA.923638, 2020.

2. Shevtsova, I., Kruse, S., Herzschuh, U., Schulte, L., Brieger, F., Stuenzi, S. M., Heim, B., Troeva, E. I., Pestryakova, L.
A., Zakharov, E. S.: Foliage projective cover of 40 vegetation sites of central Chukotka from 2018, PANGAEA, https://doi.pangaea.de/10.1594/PANGAEA.923664, 2020.

3. Shevtsova, I., Kruse, S., Herzschuh, U., Schulte, L., Brieger, F., Stuenzi, S. M., Heim, B., Troeva, E. I., Pestryakova, L. A., Zakharov, E. S.: Total above-ground biomass of 39 vegetation sites of central Chukotka from 2018, PANGAEA, https://doi.pangaea.de/10.1594/PANGAEA.923719, 2020.

4. Shevtsova, I., Kruse, S., Herzschuh, U., Schulte, L., Brieger, F., Stuenzi, S. M., Heim, B., Troeva, E. I., Pestryakova, L. A., Zakharov, E. S.: Individual tree and tall shrub partial above-ground biomass of central Chukotka in 2018, PANGAEA, https://doi.pangaea.de/10.1594/PANGAEA.923784, 2020.

**Author contributions**

IS, UH and SK designed the study. SK, IS, UH, LS, SS, LP, EZ collected the field data. SK and IS processed the field samples.
BH advised the processing of remote sensing data. IS developed R code for processing all data used in the study, performed the formal analyses and visualisation, and prepared and edited the original manuscript. SK, UH and BH supervised the research activity and provided critical review during manuscript preparation. UH, LP and SK were responsible for the management and coordination of the planning and execution of the expedition project.

**Competing interests**

The authors declare that they have no conflict of interest.

**Acknowledgements**

This study has been supported by the German Federal Ministry of Education and Research (BMBF), which enabled the Russian-German research programme "Kohlenstoff im Permafrost KoPf" (grant no. 03F0764A), by the Initiative and





Networking Fund of the Helmholtz Association and by the ERC consolidator grant Glacial Legacy of Ulrike Herzschuh (grant
no. 772852), by Russian Foundation for Basic Research (grant no. 18-45-140053 r_a), Ministry of Science and Higher
Education of the Russian Federation (grant no. FSRG-2020-0019). Birgit Heim acknowledges funding by the Helmholtz
Association Climate Initiative REKLIM. We thank our Russian and German colleagues from the joint German-Russian
expedition "Chukotka 2018" for support in the field.

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




**Figure 1: Overview of the study region and four focus areas: tundra (16-KP-04), northern tundra–taiga (16-KP-01), southern tundra–taiga (16-KP-03), and northern taiga (16-KP-02), and two areas with supplementary ABG sampling: 18-BIL-01 and 18-BIL-02 (tundra–taiga to northern taiga). Sample plot names of the 2016 expedition are V01-V58, sample plot names of the 2018 expedition are EN01-EN55 (abbreviated here to EN# rather than EN18#). Overview map modified from Shevtsova et al, 2020a. Base maps of study areas are Landsat-8 RGB composites. Black colour represents no data or water.**





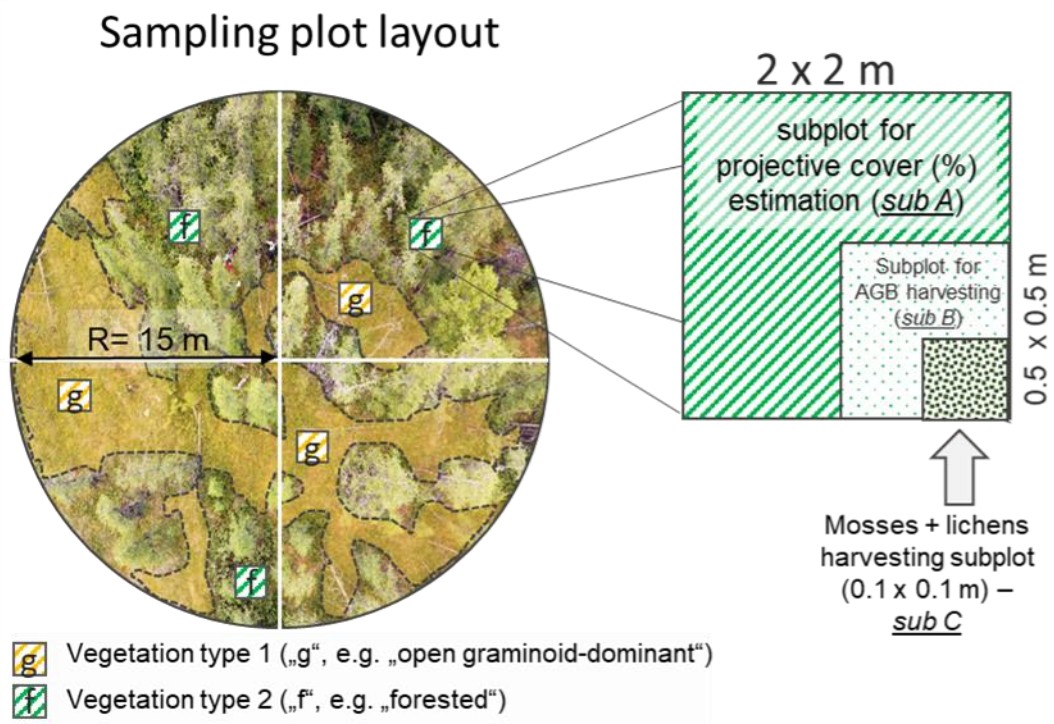

**Figure 2: Sampling scheme of the 2018 expedition vegetation survey. To accommodate heterogeneity in the main sample plot with a radius of 15 m, two to three dominant vegetation types were roughly estimated, e.g. in the example we identified two types ('g' and 'f'). Within every vegetation type, three sampling subplots (sub A, 2 x 2 m) were placed for projective cover assessment. Inside one of these, the most representative subplot per vegetation type, we placed a subplot (sub B, 0.5 x 0.5 m) for harvesting above-ground biomass (AGB) from the ground-layer plants, excluding mosses and lichens, which were instead sampled from a representative smaller subplot (sub C, 0.1 x 0.1 m).**





**Table 1: The four focus areas with corner coordinates (decimal degrees (DD), WGS 84) and acquisition times of the historical and recent Landsat spectral indices (NDVI and NDWI for peak summer, NDSI for snow-covered conditions) used for the redundancy analysis (RDA).**

| focus area | ecological zone/ ecotone | upper left coordinates (DD) | lower right coordinates (DD) | (historical image product) Landsat 7 ETM+ spectral indices | (recent image product) Landsat 8 OLI spectral indices |
|---|---|---|---|---|---|
| 16-KP-01 | northern tundra-taiga | 67.226 N, 168.096 E | 67.401 N, 168.621 E | NDVI, NDWI 30 July 2001 NDSI 24 March 2001 | NDVI, NDWI 31 July 2016 NDSI 16 March 2016 |
| 16-KP-02 | northern taiga | 67.020 N, 163.432 E | 67.173 N, 163.938 E | NDVI, NDWI 8 August 2000 NDSI 22 March 2001 | NDVI, NDWI 12 August 2016 NDSI 5 March 2016 |
| 16-KP-03 | southern tundra-taiga | 65.876 N, 166.103 E | 65.998 N, 166.509 E | NDVI, NDWI 30 July 2001 NDSI 24 March 2001 | NDVI, NDWI 31 July 2016 NDSI 16 March 2016 |
| 16-KP-04 | tundra | 67.735 N, 168.587 E | 67.831 N, 168.862 E | NDVI, NDWI 9 August 2002 NDSI 24 March 2001 | NDVI, NDWI 10 August 2017 NDSI 16 March 2016 |


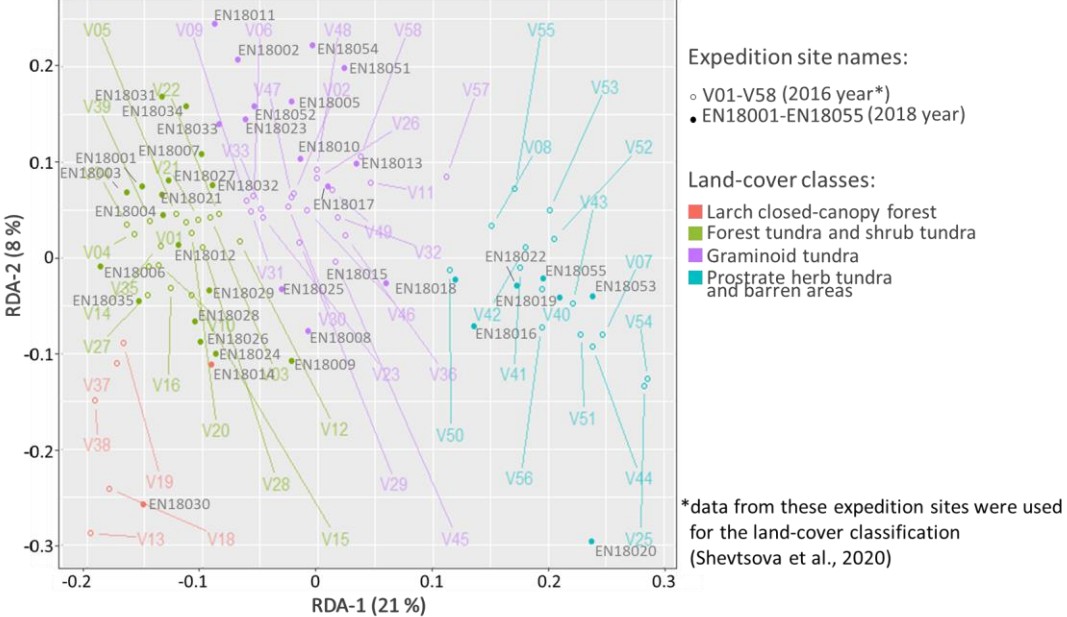

**Figure 3: 2018 expedition vegetation data predicted into RDA-space built using the 2016 expedition vegetation data and assigned to four land-cover classes: (1) larch closed-canopy forest, (2) forest tundra and shrub tundra, (3) graminoid tundra, and (4) prostrate herb tundra and barren areas.**






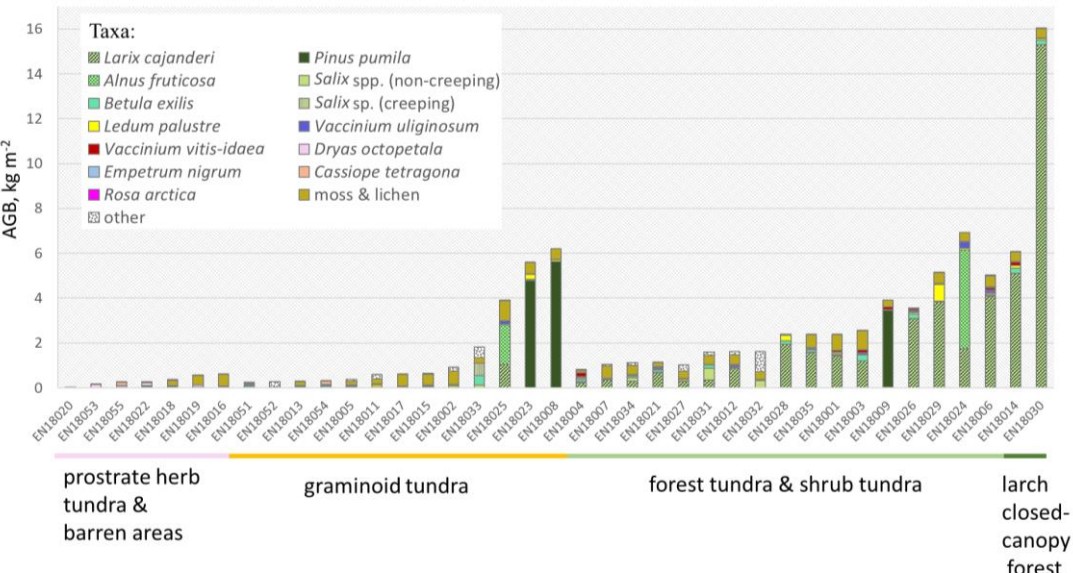

**Figure 4: In situ above-ground biomass (AGB) in kg m⁻² in each investigated sample plot according to the taxa present, ordered by the predicted land-cover class (below names of the sample plots).**

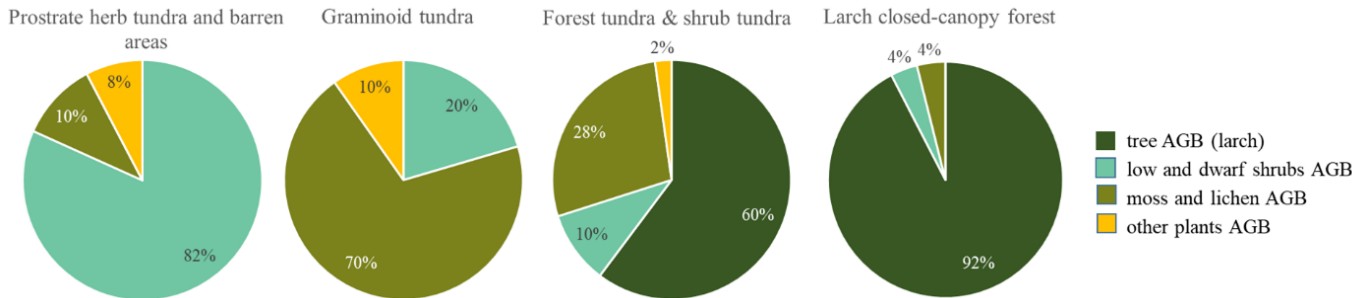


**Figure 5: Plot-scale average (median) partial AGB in the four vegetation classes. Tall shrubs (*Alnus fruticosa*, *Pinus pumila*) were rare and made up less than 1% of the average plot AGB and are not included here.**





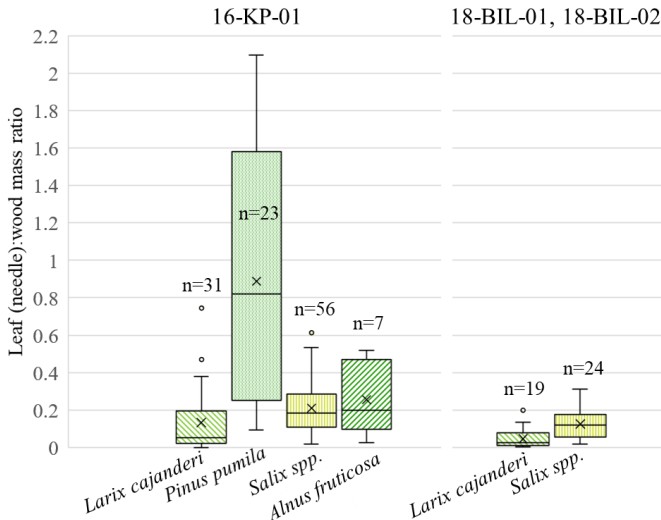

**Figure 6: Distribution of leaf (needle) to wood dry mass ratio among studied species:** *Larix cajanderi*, *Pinus pumila*, *Salix* **spp. (non-creeping), and** *Alnus fruticosa* **in two ecological regions: tundra–taiga ecotone (16-KP-01) and northern taiga (18-BIL-01, 18-BIL-02); "n" is number of individuals sampled.**

**Table 2: Estimates and significance values of generalised additive model (GAM) parameters.**

| Formula: total AGB ~ RDA1 + s(RDA1, RDA2) | | | | |
|---|---|---|---|---|
| Parametric coefficients: | | | | |
| | Estimate | Standard error | t value | p |
| (Intercept) | 2.30 | 0.20 | 11.32 | <0.005 |
| RDA1 | -0.42 | 0.06 | -6.84 | <0.005 |
| Approximate significance of smooth terms: | | | | |
| | estimated degrees of freedom | F-value | | p |
| s(RDA1,RDA2) | 10.53 | 12.04 | | <0.005 |



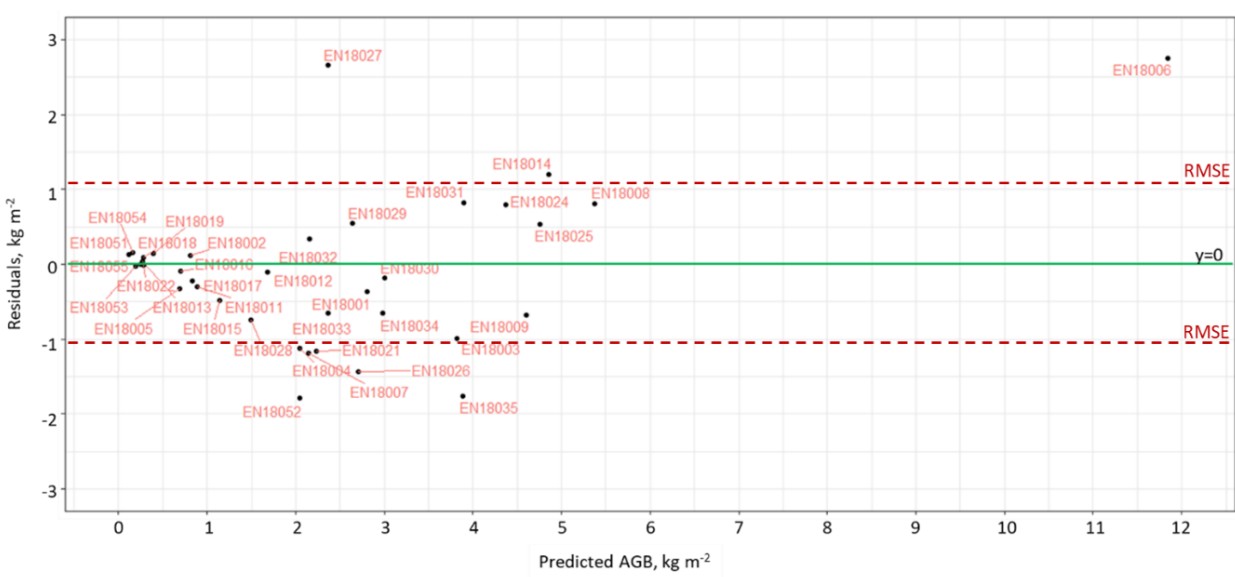

**Figure 7: The distribution of residuals of the generalised additive model (GAM) trained for AGB biomass prediction.**

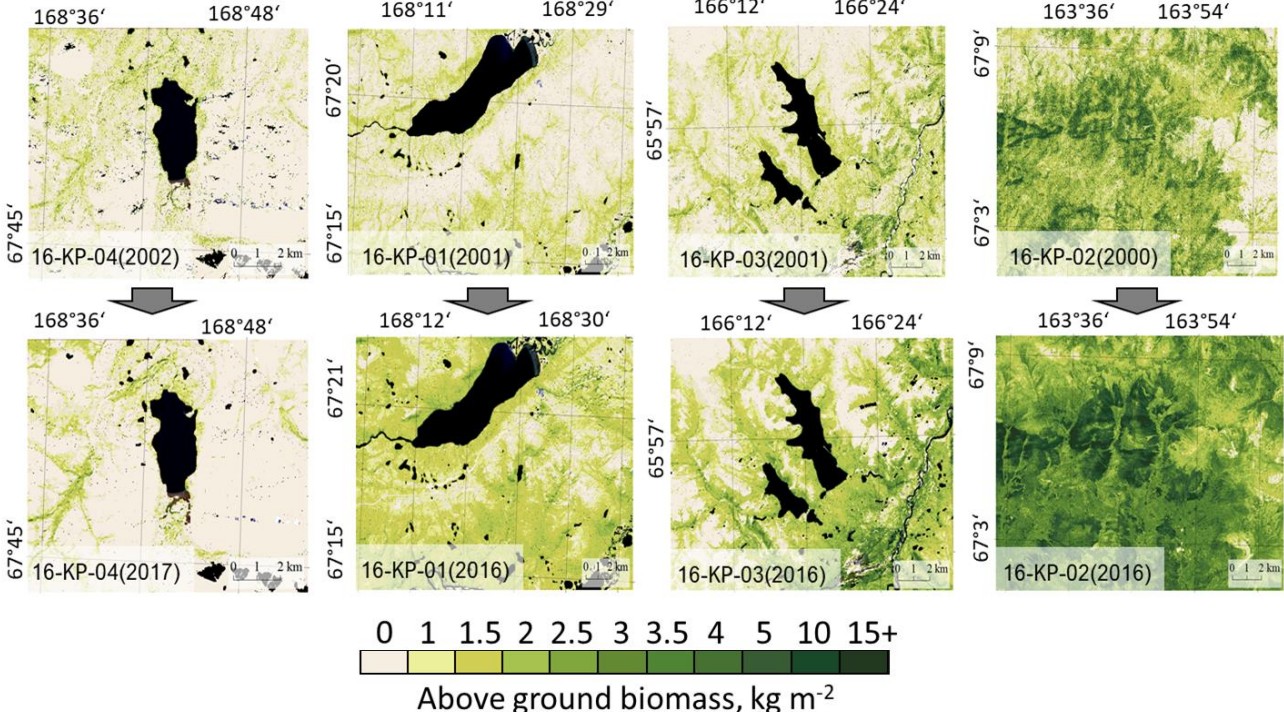

**Figure 8: Landsat-derived maps of total above-ground biomass (AGB) in historical years (2000, 2001 or 2002) and recent years (2016 or 2017) in four focus areas: treeless tundra (16-KP-04), northern tundra–taiga (16-KP-01), southern tundra–taiga (16-KP-03) and northern taiga (16-KP-770 02).**





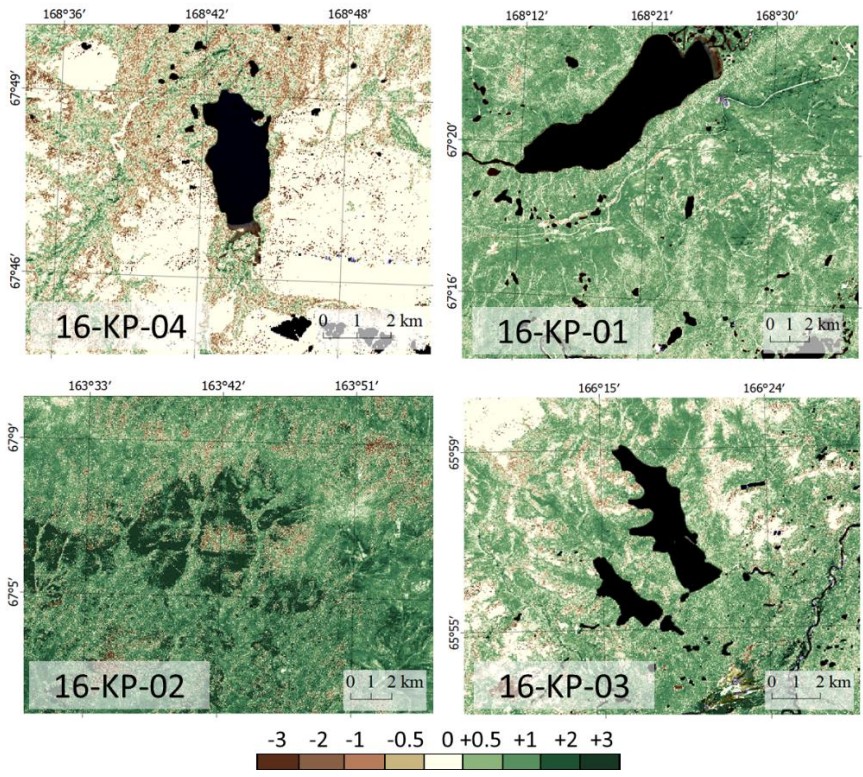

**Figure 9: Maps of change in Landsat-derived total above ground biomass (AGB) from historical years (2000/2001/2002) to recent years (20016/2017) in the four focus areas: treeless tundra (16-KP-04), northern tundra–taiga (16-KP-01), southern tundra–taiga (16-KP-03), and northern taiga (16-KP-02). A generally positive trend in AGB change is detected in the tundra–taiga and northern taiga, whereas AGB in the tundra largely remains stable or is decreasing.**



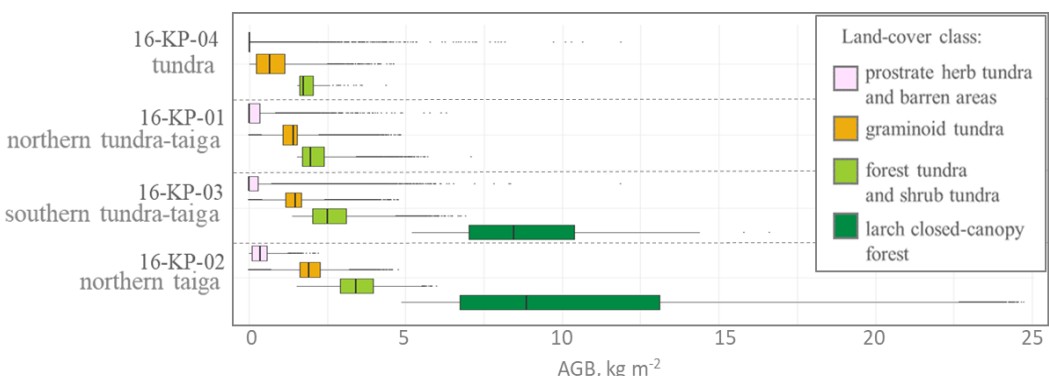


**Figure 10: Average above ground biomass (AGB) in recent years (2016/2017) within land-cover classes that have not changed between 2000 and 2017 for four investigated locations, covering a vegetation gradient from tundra (16-KP-04) via tundra–taiga (16-KP-01, 16-KP-03) to northern taiga (16-KP-02).**

**Table 3: Above-ground biomass (AGB) change associated with land-cover class change in four focus areas from 2000/2001/2002 to 2016/2017.**

| Land-cover class change | Tundra 16-KP-04 (kg m$^{-2}$) | Northern tundra–taiga 16-KP-01, (kg m$^{-2}$) | Southern tundra–taiga 16-KP-03, (kg m$^{-2}$) | Northern taiga 16-KP-02, (kg m$^{-2}$) |
|---|---|---|---|---|
| Prostrate herb tundra and barren areas -> graminoid tundra | -0.30 (IQR=0.80) | +0.20 (IQR=0.54) | +0.35 (IQR=0.95) | +1.31 (IQR=0.98) |
| Graminoid tundra -> forest tundra and shrub tundra | +0.34 (IQR=0.67) | +0.51 (IQR=0.60) | +0.65 (IQR=0.76) | +1.46 (IQR=1.04) |
| Forest tundra and shrub tundra -> larch closed-canopy forest | - | - | +4.30 (IQR=2.55) | +4.09 (IQR=3.99) |