# Peer review of "Recent above-ground biomass changes in central Chukotka (Russian Far East) using field sampling and Landsat satellite data"

_Biogeosciences, 2020_

## Referee Comment (RC1) · Anonymous Referee #1 · 9 Dec 2020

The research results presented by Shevtsova et al in the manuscript "Recent above-ground biomass changes in central Chukotka (Russian Far East) using field sampling and Landsat satellite data " make a great contribution to understanding the response of terrestrial ecosystems to climate changes in a relatively poorly studied region of the Subarctic, in particular, on the level of biomass accumulation of plant communities. Although the methods and approaches used in the collection of materials and their analysis in the study of such phenomena are valid in general, but additional clarifications are required in some points before manuscript will be published finally. Thus, on lines 94-95 it was written and on Fig.2 was demonstrated that each sample plot is round and has radius 15 m, but in Appendix A it was written on lines 468, 487, 531,

580, 609 calculations were done " for a 30 x 30 m sample plot ". In the case of round plot with 15 m radius, area of sample plot is 706,5 m2 (=3.14*15*15), but in case of a 30 x 30 m square plot, it is equal 900 m2, and the last value was used for calculation in Appendix A on lines 485-486 (A5). Please, clarify for what area was done calculation. Also it is not entirely clear how the biomass of all branches and foliage was calculated, since it is not said in lines 117-123 (for trees) or 127-129 (for shrubs), or in Appendix A, how the number of branches (from small to large) in model trees and bushes were estimated - by eye or by direct counting after they have been felled. On line 124 it was indicated that "exponential models" were used, but does not indicate where these models are presented. It should be noted that they are also listed in Appendix A (A29-A34) and it will be great if some relations between biomass of wood or needles and tree height (or stem diameter) will be demonstrated in a form of graph (figure). On lines 115-116 it was written that "heights of all trees were visually estimated in the 15 m radius plot after training with a clinometer (SUUNTO, Finland)". As a rule, the data obtained in this way have a large error (+/- several meters) if the trees are tall (more than 15 m), and the stand is dense. Such an error does not allow to accurately calculate the biomass of each tree (i.e. +/- 5-10%) on the sample plot. It was better to measure the height with a measuring rod (the average error in the presence of experience is 0.2-0.4 m) or to use for calculations also the diameter (perimeter) of the trunk, specially measured for that in the field at the stem base or on height 130 cm. The error would then decrease to 1-2%. On lines 140-143, in order to understand how a developed "redundancy analysis (RDA) model", an unprepared reader needs to familiarize himself with a significant part of the article "Shevtsova et al., 2020a". This is inconvenient for any reader, so I propose to reveal in more detail the essence of the method for developing this model in the text of the manuscript or at least in the Appendix. On lines 327-328 it was written that "stem diameter measurements (stem perimeter) were not available for all trees". What does it mean ? No measurements were taken or was it difficult to do it for some reason? It is well known that larch trunks are usually well accessible for measuring their diameters. On line 329 - I agree with

the statement that the diameter and height of trees usually correlate well with each other. And if such data were obtained, it will be great to demonstrate them in Appendix. Also, in order to be sure of the accuracy of the measurements of the biomass of trees through their height, it would be good to compare the total trees AGB calculated using the height and diameter. This is also connected with the fact that the authors of the manuscript themselves in the "Discussion" indicate some discrepancies with the data of other researchers, and therefore, in order to prove the accuracy of their calculations, such a comparison must be made. In general, in my opinion, scientific results and conclusions presented in a clear, concise, and well structured way, but in Fig. 1, it is desirable to change the color of the line denoting treeline to a more distinguishable one against the main green background, for example, red.

Please also note the supplement to this comment:
https://bg.copernicus.org/preprints/bg-2020-416/bg-2020-416-RC1-supplement.pdf

---

## Referee Comment (RC2) · Anonymous Referee #2 · 17 Dec 2020

This study used ordination to relate field-based taxa projective cover and Landsat-derived vegetation indices to upscale plant biomass distribution and dynamics. The findings indicated a general increase in total AGB throughout the investigated tundra-taiga and northern taiga, whereas the tundra showed no evidence of change in AGB. I find the results are interested, methodologies are well presented, thus believe this study is worth pubulishing. However, I still find some statements are not easy to follow and some dicussions are needed. (1) The authors have stated at Page 2, line 40-50 that 'a loss of specific species from one PFT can be replaced by taxa from another PFT in response to climate change even though total AGB production remains similar'. Accordingly, at Page 3, line 65-70, I suggest the authors should also add some literatures

which found that different PFT may also have similar NDVI values and caused bias estimation of biomass based on remote sensing data. (2) Page 4, line 90-95. I don't quite understand why plot numbers for different habitats are not equal. Please explain. (3) Page 4, line 95-100. is a 50x50 cm area large enough for tree samplings, at least in this region? How to avoid arbitrary sampling in a plot with 15 m radius? (4) Page 4, line 105-110. I am a little bit confused that sampling plots in different survey years are not in the same location? If this is the case, how to study the changes in AGB if plots located differently? The authors should provide information or cite papers to suggest to what extent these results are convincing base on such kind of data series? (5) Page 6, line 140-145. It would be better if the authors provided more information about remote sensing images used in this study even though you have cited a paper here, especially for the year of 2018. (6) Page 9, line 225-230. Maybe I missed some important information, but I did not find season information from the context. I assumed that the authors aware that when studing biomass changes, same season should be the prerequisite.

---

## Author Comment (AC1) · 15 Mar 2021

We thank the anonymous referee #1 for the revision, which helped us to clarify the outcome of our study for the reader.

Referees comment: "lines 94-95 it was written and on Fig.2 was demonstrated that each sample plot is round and has radius 15 m, but in Appendix A it was written on lines 468, 487, 531,580, 609 calculations were done " for a 30 x 30 m sample plot ". In the case of round plot with 15 m radius, area of sample plot is 706,5 m2 (=3.14*15*15), but in case of a30 x 30 m square plot, it is equal 900 m2, and the last value was used

for calculation in Appendix A on lines 485-486 (A5). Please, clarify for what area was done calculation."

Authors' response: We thank the referee for detecting some inconsistencies in our description on the sampling. We made the description in the main text more clear, edited Figure 2, and the figure text caption of Figure 2 and the text in the Appendix A accordingly. The ground layer vegetation was estimated on 2 x 2 m subplots as we wrote in section "Materials and methods" (lines 102-104). It was later recalculated in kg per square meter for the 30 x 30 m vegetation plot according to the percentage of the vegetation types' distribution on the sample plot. Tall shrubs and trees were assessed on the 15 m radius plot and recalculated in kg per square meter. Only one plot we studied more detailed and, therefore, set up a grid with a 30x30 m size also for tree AGB estimation. Thus, even if we base our estimations on the 30 x 30 m plot or 15-radius plot at the end both were recalculated into kg per square meter, which makes the differences in the area insignificant (independent) to the result.

We changed Figure 2 by showing the 30 m x 30 m vegetation plot with inside the 15 m radius circle.

We changed the figure caption of Figure 2 to (lines 807-813): "Figure 2: Sampling scheme of the 2018 expedition vegetation survey. Projective cover of tall shrubs and trees was estimated on a circular sample plot with a radius of 15 m, while ground vegetation type cover on a rectangular 30 x 30 m sample plot. To accommodate heterogeneity in the main 30 x 30 m sample plot, the two to three dominant vegetation types were estimated, e.g. in this example we identified two types ('g' and 'f'). Within every vegetation type, three sampling subplots (sub A, 2 x 2 m) were placed for projective cover assessment. Inside one of these, the most representative subplot per vegetation type, we placed a subplot (sub B, 0.5 x 0.5 m) for harvesting above-ground biomass (AGB) from the ground-layer plants, excluding mosses and lichens, which were instead sampled from a representative smaller subplot (sub C, 0.1 x 0.1 m)."

We added to the Appendix A (lines 479-481): "The ground layer vegetation AGB was estimated on a 30 x 30 m sample plot. The size and shape of the main plot were chosen to cover representative areas of present vegetation types. In contrast, the AGB of tall shrubs and trees was estimated on a 15 m radius sample plot. The final estimations were given in kg m-2, which makes them independent from their sample plot size. "

Referees comment: "Also it is not entirely clear how the biomass of all branches and foliage was calculated, since it is not said in lines 117-123 (for trees) or 127-129 (for shrubs), or in Appendix A, how the number of branches (from small to large) in model trees and bushes were estimated - by eye or by direct counting after they have been felled."

Authors' response: We added a clarification (lines 129-131): "We estimated the number of branches on each felled tree before felling by eye as following: (1) number of big branches, (2) number of medium branches on a representative big branch, (3) number of small branches on the representative medium branch. "

Referees comment: "On line 124 it was indicated that "exponential models" were used, but does not indicate where these models are presented. It should be noted that they are also listed in Appendix A (A29-A34) and it will be great if some relations between biomass of wood or needles and tree height (or stem diameter) will be demonstrated in a form of graph (figure)."

Authors' response: We added the suggested graphs (Fig. A2) into the Appendix A (lines 629-632): Figure A2: Allometric models, established for larch AGB: a - for a needle biomass of a living tree in the area 16-KP-01, b - for a wood biomass of a living tree in the area 16-KP-01, a- for a needle biomass of a living tree in the area 18-BIL, b - for a wood biomass of a living tree in the area 18-BIL, e – for a wood biomass of a dead tree in both areas.

Referees comments: "On lines 115-116 it was written that "heights of all trees were visually estimated in the15 m radius plot after training with a clinometer (SUUNTO,

Finland)". As a rule, the data obtained in this way have a large error (+/- several meters) if the trees are tall (more than 15 m), and the stand is dense. Such an error does not allow to accurately calculate the biomass of each tree (i.e. +/- 5-10%) on the sample plot. It was better to measure the height with a measuring rod (the average error in the presence of experience is 0.2-0.4 m) or to use for calculations also the diameter (perimeter) of the trunk, specially measured for that in the field at the stem base or on height 130 cm. The error would then decrease to 1-2%."

"On line 329 - I agree with the statement that the diameter and height of trees usually correlate well with each other. And if such data were obtained, it will be great to demonstrate them in Appendix."

Authors' response: Thank you for your valuable suggestions. Tree heights in our study area most of the trees were below 15 m with two exceptions (15 and 19.5 m) from 2473 trees with heights estimated. The tree stands on most of the visited forested sites were sparse and crowns of individual trees clearly seen from ground for height estimation. The tree diameter was highly correlated with tree height. We add a figure to highlight it. We added Figure A3 and a sentence in the lines 596-597: "We did not use the tree stem diameter or perimeter for this purpose, because it is highly correlated with tree height (Fig. A3). "

We added the Fig. A3 (lines 634-635): Figure A3: Relationship between tree height and perimeter of the tree stem at 0 m (a) and 1.3 m (b).

Referees comment: "On lines 140-143, in order to understand how a developed "redundancy analysis (RDA) model", an unprepared reader needs to familiarize himself with a significant part of the article "Shevtsova et al., 2020a".This is inconvenient for any reader, so I propose to reveal in more detail the essence of the method for developing this model in the text of the manuscript or at least in the Appendix."

Authors' response: We added the information into the Appendix B as it was suggested (lines 647-676):

Appendix B. Landsat data and statistical analysis of it as preparation for the AGB up-scaling

For each time stamp (2000/2001/20002 and 2016/2017) we used available Landsat acquisitions: peak-summer and snow-covered (table B1, Shevtsova et al, 2020a). We used peak-summer acquisitions to derive two Landsat spectral Indices (Normalised Difference Vegetation Index (NDVI), Normalised Difference Water Index (NDWI)) and snow-covered acquisition for derivation of Normalised Difference Snow Index (NDSI). Before indices calculation the Landsat data was topographically corrected. The subsets that we used for land-cover classification were cloud free and cloud-shadow free. Additionally, we masked all water bodies. Latdsat-8 data were transformed to Landsat-7-like (see section 1.2 Landsat data, pre-processing and spectral indices processing).

Table B1. Dates and short description of Landsat data used for retrieving spectral indices and further land-cover classification.

Landsat spectral indices NDVI, NDWI and NDSI and projective cover of different taxa were used in the RDA analysis, which made it possible to distinguish two RDA axes, which in total described 29% of the variance in the projective cover through the Landsat spectral Indices (Fig. B1).

Figure B1: The positions of the major taxa in the RDA space, based on foliage projective cover data of the plot taxa and Landsat spectral indices (Normalised Difference Vegetation Index (NDVI), Normalised Difference Water Index (NDWI) and Normalised Difference Snow Index (NDSI)), where V01-V58 are the 52 vegetation field sites (Shevtsova et al, 2020a).

Based on RDA scores we build a classification using k-means method. We were able to derive four stable land-cover classes: 1) larch closed-canopy forest, 2) forest tundra and shrub tundra, 3) graminoid tundra, 4) prostrate herb tundra and barren areas (Fig. B2).

Figure B2: K-means classes based on two redundancy analysis (RDA) axes using Normalised Difference Vegetation Index (NDVI), Normalised Difference Water Index (NDWI) and Normalised Difference Snow Index (NDSI) as predictors. Images: extracts from 360x180 degree panoramic images, Stefan Kruse.

Referees comment: "On lines 327-328 it was written that "stem diameter measurements (stem perimeter) were not available for all trees". What does it mean? No measurements were taken or was it difficult to do it for some reason? It is well known that larch trunks are usually well accessible for measuring their diameters."

Authors' response: In this case we meant, that not for all trees measurements were taken. We were not able to record tree stem perimeter for all trees due to time constrain. Following the aim of our analysis we planned to conduct a vegetation survey, having more sampling plots for use in upscaling, rather than detailed description of every tree parameters.

Referees comment: "in order to be sure of the accuracy of the measurements of the biomass of trees through their height, it would be good to compare the total trees AGB calculated using the height and diameter. This is also connected with the fact that the authors of the manuscript themselves in the "Discussion" indicate some discrepancies with the data of other researchers, and therefore, in order to prove the accuracy of their calculations, such a comparison must be made."

Authors' response: The diameter of the trees was measured only on 9 to 12 individuals per each sample plot. Height, on the other hand, was estimated for each tree on each sample plot. Therefore, there was no sufficient data to use both height and diameter in the estimations of the tree biomass. However, from available measurements we see that tree stem diameter and tree height are highly correlated (Appendix A, Fig. A3). For the purpose of demonstration, we calculated tree AGB also from both parameters and compared models. We added the results of comparison to the Appendix A (lines 637-645): "Although tree stem perimeter and tree height are closely correlated (Fig. A3),

we tested three models for reconstruction of total larch AGB to show, that tree height is enough for this purpose (table A1). The highest R2 adj=0.601 is characterising model that uses only tree height as a predictor. Lower it is for the model with both tree height and tree stem perimeter at a breast height (R2 adj=0.597), whereas tree stem perimeter is an insignificant predictor. The lowest R2 adj=0.551 is for the model that uses only tree stem perimeter at a breast height as a predictor of total tree AGB.

Referees comment: "in Fig. 1, it is desirable to change the color of the line denoting treeline to a more distinguishable one against the main green background, for example, red."

Authors' response: Thank you for the suggestion. We have changed the colour of the treeline accordingly (line 800).

Please also note the supplement to this comment:
https://bg.copernicus.org/preprints/bg-2020-416/bg-2020-416-AC1-supplement.pdf

**Supplement:**

[Figure]

**Figure A2: Allometric models, established for larch AGB: a- for a needle biomass of a living tree in the area 16-KP-01, b - for a wood biomass of a living tree in the area 16-KP-01, a- for a needle biomass of a living tree in the area 18-BIL, b - for a wood biomass of a living tree in the area 18-BIL, e – for a wood biomass of a dead tree in both areas.**

[Figure]

**Figure A3: Relationship between tree height and perimeter of the tree stem at 0 m (a) and 1.3 m (b).**

[Figure]

**Figure B1: The positions of the major taxa in the RDA space, based on foliage projective cover data of the plot taxa and Landsat spectral indices (Normalised Difference Vegetation Index (NDVI), Normalised Difference Water Index (NDWI) and Normalised Difference Snow Index (NDSI)), where V01-V58 are the 52 vegetation field sites (Shevtsova et al, 2020a).**

[Figure]

**Figure B2: *K*-means classes based on two redundancy analysis (RDA) axes using Normalised Difference Vegetation Index (NDVI), Normalised Difference Water Index (NDWI) and Normalised Difference Snow Index (NDSI) as predictors. Images: extracts from 360x180 degree panoramic images, Stefan Kruse.**

[Figure]

**Figure 1: Overview of the study region and four focus areas: tundra (16-KP-04), northern tundra–taiga (16-KP-01), southern tundra–taiga (16-KP-03), and northern taiga (16-KP-02), and two areas with supplementary AGB sampling: 18-BIL-01 and 18-BIL-02 (tundra–taiga to northern taiga). Sample plot names of the 2016 expedition are V01-V58, sample plot names of the 2018 expedition are EN01-EN55 (abbreviated here to EN# rather than EN18#). Overview map modified from Shevtsova et al, 2020a. Base maps of study areas are Landsat-8 RGB composites. Black colour represents no data or water.**

**Table A1. Statistics of the models for reconstruction total tree (larch) AGB.**

| Model formula | R² adj | Estimate | Standard error | t value | Pr (>\|t\|) |
|---|---|---|---|---|---|
| $\log(TTAGB)=a*H + b* \sqrt{BrPer1.3}+Int$ | 0.597 | a=0.004 | 0.001 | 2.977 | 0.004 |
| | | b=0.130 | 0.233 | 0.56 | 0.579 |
| | | Int=5.381 | 0.417 | 12.91 | < 2e-16 |
| $\log(TTAGB)=a*H +Int$ | 0.601 | a=0.005 | 0.0005 | 10.31 | 1.31e-15 |
| | | Int=5.527 | 0.323 | 17.11 | < 2e-16 |
| $\log(TTAGB)= b*\sqrt{BrPer1.3}+Int$ | 0.551 | a=0.78 | 0.084 | 9.31 | 7.99e-14 |
| | | Int=4.93 | 0.410 | 12.02 | < 2e-16 |

where TTAGB - total tree AGB, H - tree height,
BrPer1.3 - perimeter of tree stem at breast height or 1.3 m, a,b – coefficients, Int – intercept"

**Table B1. Dates and short description of Landsat data used for retrieving spectral indices and further land-cover classification.**

| Focus area | Landsat acquisition | | | Short description |
|---|---|---|---|---|
| | year | Month | day | (season/ Landsat mission/ spatial resolution) |
| 16-KP-01 | 2001 | 7 | 30 | peak-summer, Landsat-7, 30 m |
| | 2001 | 3 | 24 | snow-covered, Landsat-7, 30 m |
| | 2016 | 7 | 31 | peak-summer, Landsat-8, 30 m |
| | 2016 | 3 | 16 | snow-covered, Landsat-8, 30 m |
| 16-KP-02 | 2000 | 8 | 8 | peak-summer, Landsat-7, 30 m |
| | 2001 | 3 | 22 | snow-covered, Landsat-7, 30 m |
| | 2016 | 8 | 12 | peak-summer, Landsat-8, 30 m |
| | 2016 | 3 | 5 | snow-covered, Landsat-8, 30 m |
| 16-KP-03 | 2001 | 7 | 30 | peak-summer, Landsat-7, 30 m |
| | 2001 | 3 | 24 | snow-covered, Landsat-7, 30 m |
| | 2016 | 7 | 31 | peak-summer, Landsat-8, 30 m |
| | 2016 | 3 | 16 | snow-covered, Landsat-8, 30 m |
| 16-KP-04 | 2002 | 8 | 9 | peak-summer, Landsat-7, 30 m |
| | 2001 | 3 | 24 | snow-covered, Landsat-7, 30 m |
| | 2017 | 8 | 10 | peak-summer, Landsat-8, 30 m |
| | 2016 | 3 | 16 | snow-covered, Landsat-8, 30 m |

---

## Author Comment (AC2) · 15 Mar 2021

We thank the anonymous referee #2 for the revision and valuable suggestions on the improvement of our manuscript.

Referees comment: "The authors have stated at Page 2, line 40-50 that 'a loss of specific species from one PFT can be replaced by taxa from another PFT in response to climate change even though total AGB production remains similar'. Accordingly, at Page 3, line 65-70, I suggest the authors should also add some literatures which found that different PFT may also have similar NDVI values and caused bias estimation of

biomass based on remote sensing data."

Authors' response: Thank you for the suggestion. Although, there is not many publications, featuring the described effect, we added more information to make a point you were suggesting to include (lines 72-75):

"However, NDVI can be affected by water content and tall vegetation shadows, what can influence the spectral signal of vegetated land (Pattison et al, 2015) and decouple it from the biomass relationship. Such decoupling, or similar biomass ranges make distinguishing between different plant functional types (PFT) or communities difficult. Furthermore, NDVI may not capture differences in understory of moderately closed forests (Loranty at al, 2018) because the remote sensing signal comes from the top of canopy. "

Referees comment: "Page 4, line 90-95. I don't quite understand why plot numbers for different habitats are not equal. Please explain."

Authors' response: Before the expedition to the previously not described in terms of vegetation central Chukotka we planned to cover different habitats based on NDVI. On the other hand, afterwards we based our vegetation classification on taxonomical composition, rather than NDVI, what mainly made the disproportions in the sampling different habitats. However, the different number of habitats is in line with the concept of stratified random sampling, assuming higher number of plots to place the well-presented typical habitats and less in the not typical. We added a clarification to the lines 98-100: "Numbers of plots per habitat are different, but align with the concept of stratified random sampling with assuming a higher number of plots to place the well-presented typical habitats and less in the not typical."

Referees comment: "Page 4, line 95-100 is a 50 x 50 cm area large enough for tree samplings, at least in this region? How to avoid arbitrary sampling in a plot with 15 m radius?"

Authors' response: The trees were not sampled on the 50 x 50cm area, this area was used only to sample ground layer vegetation. The trees were sampled on the plot with 15-m radius. We added a clarification (line 106): "Trees and tall shrubs were sampled directly from 15 m radius plot."

Referees comment: "Page4, line 105-110. I am a little bit confused that sampling plots in different survey years are not in the same location? If this is the case, how to study the changes in AGB if plots located differently? The authors should provide information or cite papers to suggest to what extent these results are convincing base on such kind of data series?"

Authors' response: Speaking of AGB changes we only compared changes inferred from Landsat satellite data. We used field AGB estimations of 2018 to establish a connection between the field and remote sensing data. We have not compared field-based AGB changes in different years. For the clarification, we have added a sentence (lines 114-115): "In 2016, we investigated only projective cover, whereas in 2018 both projective cover and AGB were estimated."

Referees comment: "Page 6, line 140-145. It would be better if the authors provided more information about remote sensing images used in this study even though you have cited a paper here, especially for the year of 2018."

Authors' response: We used the remote sensing data from 2000/2001/2002 and 2016/2017. For clarification we added the description of remote sensing data used in the study in Appendix B (lines 647-659):

"For each time stamp (2000/2001/2002 and 2016/2017) we used available Landsat acquisitions: peak-summer and snow-covered (table B1, Shevtsova et al, 2020a). We used peak-summer acquisitions to derive two Landsat spectral Indices (Normalised Difference Vegetation Index (NDVI), Normalised Difference Water Index (NDWI)) and snow-covered acquisition for derivation of Normalised Difference Snow Index (NDSI). Before indices calculation the Landsat data was topographically corrected. The subsets that we used for land-cover classification were cloud free and cloud-shadow free. Additionally, we masked all water bodies. Latdsat-8 data were transformed to Landsat-7-like (see section 1.2 Landsat data, pre-processing and spectral indices processing).

Table B1. Dates and short description of Landsat data used for retrieving spectral indices and further land-cover classification.

Referees comment: "Page 9, line 225-230. Maybe I missed some important information, but I did not find season information from the context. I assumed that the authors aware that when studding biomass changes, same season should be the prerequisite."

Authors' response: In the line 96 we stated that "during the expedition "Chukotka 2018" in July 2018..." the survey in the field was done in July. Concerning Landsat data we used the peak-vegetation season (15 July-15 August). That information was added to the Appendix B (table B1).

Please also note the supplement to this comment:
https://bg.copernicus.org/preprints/bg-2020-416/bg-2020-416-AC2-supplement.pdf

**Supplement:**

**Table B1. Dates and short description of Landsat data used for retrieving spectral indices and further land-cover classification.**

| Focus area | Landsat acquisition | | | Short description |
|---|---|---|---|---|
| | year | Month | day | (season/ Landsat mission/ spatial resolution) |
| 16-KP-01 | 2001 | 7 | 30 | peak-summer, Landsat-7, 30 m |
| | 2001 | 3 | 24 | snow-covered, Landsat-7, 30 m |
| | 2016 | 7 | 31 | peak-summer, Landsat-8, 30 m |
| | 2016 | 3 | 16 | snow-covered, Landsat-8, 30 m |
| 16-KP-02 | 2000 | 8 | 8 | peak-summer, Landsat-7, 30 m |
| | 2001 | 3 | 22 | snow-covered, Landsat-7, 30 m |
| | 2016 | 8 | 12 | peak-summer, Landsat-8, 30 m |
| | 2016 | 3 | 5 | snow-covered, Landsat-8, 30 m |
| 16-KP-03 | 2001 | 7 | 30 | peak-summer, Landsat-7, 30 m |
| | 2001 | 3 | 24 | snow-covered, Landsat-7, 30 m |
| | 2016 | 7 | 31 | peak-summer, Landsat-8, 30 m |
| | 2016 | 3 | 16 | snow-covered, Landsat-8, 30 m |
| 16-KP-04 | 2002 | 8 | 9 | peak-summer, Landsat-7, 30 m |
| | 2001 | 3 | 24 | snow-covered, Landsat-7, 30 m |
| | 2017 | 8 | 10 | peak-summer, Landsat-8, 30 m |
| | 2016 | 3 | 16 | snow-covered, Landsat-8, 30 m |